

# Extrapolating regional probability of drying of headwater streams
# using discrete observations and gauging networks
**Aurélien BEAUFORT**[1], Nicolas LAMOUROUX[2], Hervé PELLA[2], Thibault DATRY[2] and Eric SAUQUET[1]
[1]Irstea, UR HHLY, Hydrology and Hydraulics Research Unit, 5 rue de la Doua CS 20244, 69625
Villeurbanne Cedex, France
[2]Irstea, UR MALY, Laboratory Dynam, 5 rue de la Doua CS 20244, 69625 Villeurbanne Cedex, France
## Abstract
Headwater streams represent a substantial proportion of river systems and have frequently flows
intermittence due to their upstream position in the network. These intermittent rivers and
ephemeral streams have recently seen a marked increase in interest, especially to assess the impact
of drying on aquatic ecosystems. The objective of this paper is to quantify how discrete (in space and
time) field observations of flow intermittence help to extrapolate the daily probability of drying at
the regional scale. Two empirical models based on linear or logistic regressions have been developed
to predict the daily probability of intermittence at the regional scale across France. Explanatory
variables were derived from available daily discharge and groundwater level data of a dense
gauging/piezometer network, and models were calibrated using discrete series of field observations
of flow intermittence. The robustness of the models was tested using (1) an independent, dense
regional data set of intermittence observations, (2) observations of the year 2017 excluded from the
calibration. The resulting models were used to simulate the regional probability of drying in France:
(i) over the period 2011-2017 to identify the regions most affected by flow intermittence; (ii) over the
period 1989-2017, using a reduced input dataset, to analyze temporal variability of flow
intermittence at the national level. The two regressions models performed equally well between
2011 and 2017. The accuracy of predictions depended on the number of continuous
gauging/piezometer stations and intermittence observations available to calibrate the regressions.



Regions with the highest performance were located in sedimentary plains, where the monitoring
network was dense and where the regional probability of drying was the highest. Conversely, worst
performances were obtained in mountainous regions. Finally, temporal projections (1989-2016)
suggested highest probabilities of intermittence (> 35%) in 1989-1991, 2003 and 2005. A high density
of intermittence observations improved the information provided by gauging stations and
piezometers to extrapolate the spatial distribution of intermittent rivers and ephemeral streams.
**Keywords:** Intermittent rivers, headwater streams, flow regime, discrete observations, regional scale
# 1. Introduction
Headwater streams represent a substantial proportion of river systems (Leopold et al., 1964; Nadeau
and Rains, 2007; Benstead and Leigh, 2012). From an ecological point of view, headwater catchments
are at the interface between terrestrial and aquatic ecosystems and they often harbour a unique
biodiversity with a very high spatial turn-over (Meyer et al., 2007; Clarke et al., 2008; Finn et al.,
2011). Their contribution to the functioning of hydrographic networks is essential: sediment flows,
inputs of particulate organic matter and nutrients, refugia/colonization, sources for aquatic
organisms (Meyer et al., 2007; Finn et al., 2011).
Headwater streams are generally naturally prone to flow intermittence, i.e. streams which stop
flowing or dry up at some point in time and space, mainly due to their upstream position in the
network and their high reactivity to natural or human disturbances (Benda et al., 2005; Datry et al.,
2014b). These waterways which cease flow and/or dry are referred as intermittent rivers and
ephemeral streams (IRES). The geographic extent of IRES is poorly documented due to mapping
limitations (digital elevation models, satellite images, aerial photos) and because of their size and
their location (Leopold et al., 1994; Nadeau and Rains, 2007; Benstead and Leigh, 2012; Fritz et al.,
2013). However the proportion of IRES in hydrological networks can be very large: for example, they
represents 60% of the length of rivers in the United States (Nadeau and Rains, 2007) and are



considered to represent probably more than 50% of the global hydrological network (Larned et al.,
2010; Datry et al., 2014b). Considering only gauging stations with continuous records may lead to
severely underestimate their regional extent (Snelder et al., 2013; De Girolamo et al., 2015; Eng et
al., 2016).
Recently, IRESs have seen a marked increase in interest stimulated by the challenges of water
management facing the global change context (water scarcity issues, climate change impact, etc.)
(Acuña et al., 2014; Datry et al., 2016b). Studies have characterized the hydrological functioning of
IRES (Gallart et al., 2012; Costigan et al., 2016) to assess the effects of flow intermittence on aquatic
ecosystems (Larned et al., 2010; Datry et al., 2016b; Leigh et al., 2016; Leigh and Datry, 2017). IRES
have been altered due to human actions (abstraction, hill dams, low-water support, pollution, etc.)
despite their high and unique biodiversity (Datry et al., 2014; Garcia et al., 2017a). In addition, some
perennial streams are becoming intermittent due to global change, water abstraction or river
damming (Skoulikidis, 2009) and the extent of IRES may increase in the future (Döll and Schmied,
2012; Jaeger et al., 2014; Pumo et al., 2016; Garcia et al., 2017b; De Girolamo et al., 2017).
A better hydrological understanding of IRES is now essential and an improved management requires
knowing both the spatial extent and arrangement of IRES within the river network (Boulton, 2014;
Acuña et al., 2017). Efforts have been made to estimate the spatial distribution of IRES at the
catchment scale (Skoulikidis et al., 2011; Datry et al., 2016a), at the regional scale (Gómez et al.,
2005) and at the national scale (Snelder et al., 2013). In France, Snelder et al. (2013) suggested a
classification of IRES regimes and spatialized their distribution. Based on an analysis of the
continuous gauging network, they showed that the proportion of IRES accounted for 20 to 39% of the
hydrographic network. The accuracy of the obtained map is highly dependent on the density of the
flow monitoring network. The installation of additional gauging stations is expensive and headwaters
systems may be difficult to monitor due to active geomorphology processes or to difficult access.



As a promising tool to advance the mapping of IRES, citizen science has proved to create
opportunities to overcome the lack of hydrological data and lead to densify the flow state
observation network (Turner and Richter, 2011; Buytaert et al., 2014; Datry et al., 2016b). In France,
Datry et al. (2016a) used such data to describe the spatiotemporal dynamics of aquatic and
terrestrial habitats within five river catchments located in the western part of France. They showed
that processes resulting in flow intermittence were complex at a fine scale and could vary
substantially among nearby catchments. However, these data were only available in few catchments,
limiting any attempt to map large-scale patterns of flow intermittence in river networks. Since this
first attempt, new sources of observational data have become available in France thanks to the
ONDE network (Observatoire National des Etiages, onde.eaufrance.fr). This unique network in
Europe provides frequent discrete field observations (five inspections per year) of the flow
intermittence across more than 3 300 sites throughout France and located mostly in headwater
areas.
The objective of this paper is to quantify how discrete (in space and time) field observations of flow
intermittence help to extrapolate the daily probability of drying at the regional scale. We first carried
out a quantitative analysis of the ONDE network data and to characterize their information
contribution in comparison with the data resulting from the conventional hydrological monitoring.
Then, two empirical models based on linear or logistic regressions have been developed to convert
discontinuous series of flow intermittence observation from ONDE into daily probability of drying at
the regional scale across France. Explanatory variables were derived from available daily discharge
and groundwater level data of a dense gauging/piezometer network, and models were calibrated
using the ONDE discrete observations. The robustness of the models was tested using (1) an
independent, dense regional data set of intermittence observations and (2) observations of the year
2017 excluded from the calibration. Finally, resulting models were used to simulate the regional
probability of drying in France: (i) over the period 2012-2016 to identify the regions most affected by



flow intermittence; (ii) over the period 1989-2016, using a reduced input dataset, to analyze
temporal variability of flow intermittence at the national level.

## 2. Material and Methods

### 2.1. Study area

The study area is continental France and Corsica (550 000 km²). France is located in a temperate zone
characterized by a variety of climates due to the influences of the Atlantic Ocean, the Mediterranean
Sea and mountain areas.
We defined regions as combinations of "level-2 hydroecoregions" (HER2) and classes of hydrological
regimes (HR). Hydro-EcoRegion (HER) correspond to a typology developed for river management in
accordance with the European Water Framework Directive. The Hydro-ecoregions classification
includes 22 "level-1 hydroecoregions" (HER1) based on geology, topography and climate, and
considered as the primary determinants of the functioning of water ecosystems (Wasson et al.,
2002). HER2 regions correspond to a finer classification accounting for stream size. HER2 have a
mean drainage area of 5 000 km² (between 100 and 27 000 km²). The hydrological regimes classes
(HR) were identified by reference to the work carried out by (Sauquet et al., 2008) where it was
possible to distinguish rainfall-fed regimes, transition and snowmelt-fed river flow regimes. Overall,
we used 280 regions (HER2-HR combinations) with a mean drainage area of 1 400 km² (between 4
and 20 000 km²).

### 2.2. ONDE dataset discrete national flow-state observations

The ONDE network was set up in 2012 by the French Biodiversity Agency (AFB, formerly ONEMA)
with the aim of constituting a perennial network recording summer low flow levels and used to
anticipate and manage water crisis during severe drought events (Nowak and Durozoi, 2012).





There are 3 300 ONDE sites distributed throughout France (Fig. 1). ONDE sites are located on
headwater streams with a Strahler order strictly less than 5 and balanced across HER2 regions to take
into account the representativeness of the hydrological contexts (Nowak and Durozoi, 2012). The
ONDE network is stable over time. Observations are made monthly (around the 25th) by trained AFB
staff, between April and September, every year since 2012. One of the statuses is assigned at each
observation among "visible flow", "no visible flow" and "dried out". Here, we consider two
intermittency statuses: "**Flowing**" when there is visible flow across the channel ("visible flow") and
"**Drying**" when the channel is entirely devoid of surface water ("dried out") or when there is still
water in the river bed but without visible flow (disconnected pools, lentic systems) ("no visible
flow"). The proportion of drying sites determined on the basis of the ONDE network for each HER2-
HR combination is considered as a good estimate of the daily Regional Probability of Drying
(RPoD$_{ONDE}$) of streams with a Strahler order less than 5.
Figure 2 illustrates the complementary nature of the ONDE network to the already existing French
river flow monitoring network HYDRO (http://www.hydro.eaufrance.fr/). The ONDE sites and a set of
1 600 gauging stations available in the HYDRO database have been projected on the river network
RHT (Theoretical Hydrographic Network; Pella *et al*., 2012) and the drainage area and the elevation
have been estimated. A large part of ONDE sites are located on small headwater streams with 70% of
the sites with a drainage area of less than 50 km² while most of the gauging stations record flows of
catchment of medium size (between 100 and 500 km²). Only four stations display a drainage area of
more than 1 000 km². The distributions of elevation of the two databases look similar. The ONDE
sites are mostly located on rivers with an elevation below 200 m (75% of sites). The ONDE sites are
sparse at high elevations (95 sites located above 1 000 m). The more likely reason of this bias is the
difficult access to the river beds in mountainous area.





### 2.3. POC dataset: a denser regional dataset used for independent validation

A spatially denser citizen science dataset of flow-state observations in western France (Poitou-Charente region) (http://atlas.observatoire-environnement.org) has been used as validation dataset to test the robustness of our models calibrated with the ONDE dataset. The POC monitoring (2011-2013) covered more than 4 000 km of river length across 20 catchments. Each river was entirely surveyed every 1$^{st}$ and 15$^{th}$ of each month between June and October, resulting in eight observations per year. Four intermittency statuses were available in the POC dataset (Datry *et al*. 2016a) but to allow comparisons with the ONDE network, we pooled the two "Flowing" and "Low Flow" POC statuses into a single "**Flowing**" status and the two "No flow" and "Dry" statuses into the "**Drying**" status. This dataset is available as maps with flow states assigned to the inspected streams. The proportion of drying derived from this dataset for each HER2-HR combination located in the Poitou-Charente region is hereafter called RPoD$_{POC}$.

### 2.4. Explanatory hydrological dataset

Two hydrological datasets were used as explanatory variables of discrete intermittence observations and for extrapolating the intermittence frequency over time. The two datasets included time series of daily discharge extracted from the French River discharge monitoring network ("HYDRO database", http://www.hydro.eaufrance.fr/): (i) **the 2011-2017 dataset** with full records available between the 01/01/2011 and 31/06/2017; (ii) **the 1989-2017 dataset** concerning a reduced number of gauging stations and providing daily discharges between the 01/01/1989 and 31/06/2017. According to the hydrometric services in charge of the selected gauging stations, high quality of measurements was ensured and observed discharges were not or only slightly altered by human actions.

The 2011-2017 dataset was composed by 1 600 gauging stations distributed across France. Each stream where a HYDRO gauging station is located has been defined as IRES or perennial. Several definitions of IRES can be found in the literature (Huxter and van Meerveld, 2012, Eng et al., 2016;




Reynolds et al., 2015). In this study, we considered stations as intermittent when five consecutive
days with discharge less than 1 liter per second has been observed during the period of record.
The 1989-2017 dataset consisted of 630 gauging stations selected with less than 5% of missing data
(continuous or not) during the period 1989-2017. This dataset has been thereafter used to estimate
the proportion of drying before the creation of the ONDE network.

### 2.5.   Explanatory groundwater level dataset

Because groundwater resources influence stream intermittence, we used available time series of the
daily groundwater level available in the ADES database (http://www.ades.eaufrance.fr/) at sites
identified as involved in groundwater/surface water exchanges (Brugeron et al., 2012). Similarly to
the hydrological data, two sets of groundwater level data with records available over the two periods
2011-2017 and 1989-2017 have been selected.
The 2011-2017 dataset was composed by 750 piezometers with daily groundwater level data with
less than 5% of missing data (continuous or not).
The selection of 1989-2017 dataset was not easy because few groundwater level measurements
were available in the database before 2000. For example, only five piezometers met the tolerance
limit on missing values considered for the 1989-2017 hydrological dataset. In order to extend the
dataset and because groundwater levels were less variable than stream discharges, the proportion of
permitted gaps was fixed to 20% between 1989 and 2017. This led us to select 150 piezometers.
Thereafter, when the missing data period was less than 10 days, groundwater level were
reconstructed by linear interpolation in order to reduce the proportion of missing values to less than
5% for the 150 piezometers selected.

### 2.6.   Statistical modeling of regional probability of drying

A regional probability of drying (RPoD) was calculated on each sampling date and for each HER2-HR
combination. The parametric modeling strategy was based on 4 main steps (Fig. 3). The first step





consisted in selecting all ONDE sites, gauging stations and piezometers located in a same HER2-HR
combination. When the total number of gauging stations and piezometers was less than 5 for a
HER2-HR combination, we merged 20 of the 280 regions with a neighboring one located in the same
HER1. The second step consisted in calculating the $\text{RPoD}_{ONDE}$ for each observation date (5 per year)
and for all selected ONDE sites. In a third step, a flow duration curve was determined for each
selected HYDRO gauging station. The average non-exceedance frequency of the observed discharge
at gauging stations was calculated between the date of observation (d) at ONDE sites and the 5 days
preceding the observation. The lag of six days accounted for the fact that ONDE survey dates in a
region could differ by 5 days, and accounted for the inertia of physical processes (e.g. storage
capacity); it was chosen after a few trials. The same operation was carried out with selected
piezometers. Finally the hydrological conditions are described by the weighted average F of the non-
exceedance frequencies of discharge and groundwater levels with respect to the relative proportions
of gauging stations and piezometers.
The fourth step consisted in estimating the $\text{RPoD}_{ONDE}$ as a function of F. Two types of regression were
fitted for each HER2-HR combination across France:
a truncated logarithmic linear regression (LLR), with two parameters $\alpha_1$ and $\beta_1$:

$$\text{RPoD}_{LLR} = \begin{cases} \min(1; \alpha_1 \times \ln(F) + \beta_1) \ when \ F < F0 \\ 0 \ when \ F \geq F0 \end{cases} \qquad (1)$$


F0 was fixed as the value of non-exceedance frequencies of discharge and groundwater levels at
which no more drying was observed across the ONDE network ($\text{RPoD}_{ONDE} = 0$).
a logistic regression (LR), with two parameters $\alpha_2$ and $\beta_2$:

$$Logit(\text{RPoD}_{LR}) = ln\left(\frac{\text{RPoD}_{LR}}{1-\text{RPoD}_{LR}}\right) = \alpha_2 \times F + \beta_2 \qquad (2)$$


LR is a multivariate analysis method well known for its relevance in binary classification issues (Lee,
2005). The $\text{RPoD}_{LR}$ was then calculated as following Eq. 3:





$$\mathrm{RPoD}_{LR} = \frac{\exp(\alpha_2 + \beta_2 F)}{1 + \exp(\alpha_2 + \beta_2 F)} \tag{3}$$


## 2.7. Model robustness: validation using independent data sets


We used the POC independent data and the 2017 ONDE year to test the robustness of the LLR and LR


model to predict the intermittence frequency in space and time. Note than when predicting on the


POC datasets, a new model was calibrated using only ONDE sites located out of POC streams.


For both datasets (POC and ONDE 2017), the relative performance of the LLR and LR models was


compared in multiple ways using both the 2011-2017 and the 1989-2017 datasets. The performance


of each model was evaluated by the Nash-Sutcliffe efficiency criterion (NSE) (Nash and Sutcliffe,


1970):


$$NSE = 1 - \frac{\sum_{i=1}^{N} (\mathrm{RPoD}_{\mathrm{ONDEi}} - \mathrm{RPoD}_{\mathrm{pri}})^2}{\sum_{i=1}^{N} (\mathrm{RPoD}_{\mathrm{ONDEi}} - \overline{\mathrm{RPoD}_{\mathrm{ONDEi}}})^2} \tag{4}$$


where $\mathrm{RPoD}_{\mathrm{ONDEi}}$ is the average proportion of drying over the ONDE sites located in the HER2-HR


combination at the $i^{\mathrm{th}}$ observation date, $\mathrm{RPoD}_{\mathrm{pri}}$ is the predicted regional probability of drying at the


$i^{\mathrm{th}}$ observation date, $\overline{\mathrm{RPoD}_{\mathrm{ONDEi}}}$ is the mean of $\mathrm{RPoD}_{\mathrm{ONDEi}}$ over the period and $N$ is the total number


of observations in the ONDE network for each HER2-HR combination.


## 2.8. Model prediction


Both models have been calibrating over the period 2012-2016 and were then applied in a $5^{\mathrm{th}}$ step to


predict the daily RPoD in France (Fig. 3). The RPoD was firstly predicted over the period 2012-2016 in


order to identify the most affected regions by flow intermittence using the 2011-2017 datasets. The


second application concerned the prediction of RPoD in France over a longer period using the 1989-


2017 dataset to analyze the temporal variability of flow intermittence at the national level. It should




be noted that models predictions only concern streams with a Strahler order lower than 5 due to the
ONDE sites location.

## 3. Results

### 3.1. Quantitative analysis

#### 3.1.1. Inter-annual intermittence analysis with the ONDE network

A total of 1 127 ONDE sites have recorded at least one drying during the period 2012-2016
representing 35% of the 3 300 ONDE sites. Between 2012 and 2016, the most critical year is 2012
with 15% of drying followed by 2016 (14%) and 2015 (14%) (Fig. 4a). The year 2013 and 2014 are less
affected with only 6% of drying observed (Fig. 4a).
Dryings mainly occur between July and September but the evolution of the month's proportion of
drying can differ between years (Fig. 4). In more details, water levels in 2012 decrease in August
when the proportion of drying is 27% and the situation lasts until the end of September with 25% of
drying (Fig. 4b). In 2013, the drying proportion is lower than in 2012 but follow the same pattern with
an increased at the end of July (3%) and reached 9% in August and in September. In 2014, the first
peak of drying (5%) is reached early in June. Then, the drying proportion decreases in July (3%) and
increases slightly in August 4% and reach 7% in September. In 2015, the critical period occur at the
end of July with 19% of drying and the proportion of drying decreases slightly at the end of August
(17%) until reaching 9% in September. Finally, in 2016, the situation is gradually deteriorated every
month, reaching 20% of drying in August, and 28% in September.
Between 2012 and 2016, a proportion of drying higher than 50% is recorded on 93 ONDE sites and
their spatial distribution is very patchy at the France scale (black and dark grey dots, Fig. 5a). There
are only 158 ONDE sites with at least one drying every year and a variability of drying locations can
be observed across years. The south-east of France is heavily affected by rivers drying where drying
proportion can exceed 75% annually (black dots, Fig. 5b-5f). The north-western part of France is less



affected, although many ONDE sites show a drying proportion observed above 50% in 2014 and 2016
(Fig. 5d and 5f). Northeastern France is rather affected in 2012, 2014 and 2015 where several ONDE
sites have more than 75% of drying (Fig. 5b, 5d and 5e). The south-west France is particularly
affected in 2012 and 2015 (Fig. 5b and 5e).

### 263    3.1.2.  Comparison of flow intermittence between ONDE network and HYDRO dataset

HYDRO dataset includes 90 gauging stations located on streams considered as IRES, which represents
only 5.5% of the 3 300 gauging stations against 35% for ONDE sites. At the national scale, the number
of IRES seems underrepresented in the south-western, central, northeastern part of France and
Corsica in comparison with sites experiencing drying in the ONDE network (Fig. 6).
The number of gauging stations with at least one drying (discharge < 1 l/s) observed between May
and September varies between 79 in 2012 and 47 in 2014 (Table 1). The lowest numbers of gauging
stations with drying are observed in the years 2013 and 2014 while the highest numbers are related
to the years 2012, 2015 and 2016. This finding is consistent with the analysis of the ONDE network
(Fig. 5a, d). The frequency of drying calculated between the $1^{st}$ May and $30^{st}$ September, in contrast,
is quite constant over the years (~30%). The number of gauging stations with drying over more than
50% of the time varies little between wet years (14 in 2013) and dry years (21 in 2015) unlike ONDE
observations which suggest a significant temporal variability in the frequency of drying between dry
and wet years (Fig. 5).

## 277    3.2.    Validation of the predicted regional probability of drying

### 278    3.2.1.  Regression results

LLR and LR models, calibrated over the period 2012-2016, perform well with the 2011-2017 dataset
with a mean NSE of 0.8 with LR model against 0.7 with LLR model (Fig. 7a and b). With the LR model,
50% of the HER2-HR combinations obtain a NSE greater than 0.8, representing a coverage of 65% of
the French territory, while 33% of HER2-HR combinations display a NSE higher than 0.8 (50% of
France coverage) with the LLR model. Regions with the highest performances are located in



sedimentary plains, in the south-east of France and in the Pyrenees Mountains. Conversely, the
worst performances are obtained in the mountainous regions of Alps as well as in the Massif Central.
In these regions the size of the HER2 is rather small and the number of ONDE sites, gauging stations
and piezometers per HER2-HR combinations are certainly too few to derive reliable relations. Despite
pooling, estimating RPoD remains impossible for 9 HER2-HR combinations (4.5% of France coverage)
because the number of ONDE sites, gauging stations and piezometers sites is insufficient (less than 5)
to perform the regression analysis.
The performance level is lower when the 1989-2017 dataset is used in models: the mean NSE with
the LR and LLR models is 0.7 and 0.6, respectively (Fig. 7c and d).
The LR and LLR models lead to similar performance range. However, the LR model outperforms the
LLR model in terms of number of HER2-HR combinations with NSE greater than 0.8 (Fig. 7c and d).
The performance is sensitive to the dataset. As expected, the best results are obtained with the
denser network. A decrease in NSE by more than 0.2 is identified for 5% of the French territory when
the 1989-2017 dataset is used (black areas; Fig. 7e and f). The regions with the most degraded values
of NSE are small HER2-HR combinations located in eastern France (Fig. 7e and f).
The decrease in performance is mainly due to the difference in number of gauging stations and
piezometers between the two datasets (Fig. 8). The most degraded NSEs correspond to HER2-HR
combinations where the number of gauging stations and piezometers considered in regressions is
the most reduced, i.e. with a loss higher than 50% of stations (black and dark grey dots; Fig 8a and b).
However, the decrease in performance remains low (difference in NSE is below 0.1 for 75% and 64%
of HER2-HR combinations with LLR and LR model, respectively).

### 305     3.2.2.  Comparison to the POC database

The observed proportion of drying $RPoD_{POC}$ is rather well simulated by both LLR and LR models with
the 2011-2017 dataset (NSE > 0.7 except for the year 2011, Fig. 10). In addition, the models are able
to capture small fluctuations of $RPOD_{POC}$ during the summer period. The best results during the year



2011 are obtained with the LLR model (black curve; Fig. 9) and the LR model overestimates $RPoD_{POC}$
by 3% (dashed grey curve; Fig. 9). In 2012, the decline in water levels is more gradual than in 2011
and a marked peak is reached in September with 40% of $RPoD_{POC}$ (Fig. 9). This pattern is well
reproduced by both models with a good fit to all observation points (Fig. 9). The year 2013 is less
affected by drying occurrence and the maximum $RPoD_{POC}$ does not exceed 20% (Fig. 9). Curves of
both models fit to observations well until the end of August. Note that the LR model is slightly closer
to the observations around the peak in September compared to the LLR model. However the LR
model overestimates the $RPoD_{POC}$ at the end of September and in October.
The simulations of RPoD are weakly degraded when both models use with the 1989-2017 dataset
(Fig. 9d, e, f). However the simulated pattern is similar to the observed one. The LLR model
outperforms the LR model during the three years of validation with the 1989-2017 dataset (black
curve; Fig. 9d, e, f).

### 321    3.2.3.    Temporal patterns assessment of models between 2012 and 2017

The LLR and LR models perform similarly whatever the dataset (with an average NSE of 0.7 between
2012 and 2017 with the two datasets, Tab. 2). Note that both models tend to better simulate the
RPoD during dry years 2012 and 2016 (NSE = 0.8 with LLR and LR models; Tab. 2) than during wet
years (e.g. 2014 with NSE < 0.7).
Less satisfactory performance is achieved for the two months in the year 2017 (NSE < 0.5). Figure 10
shows the dispersion between simulated RPoD and drying observed at ONDE sites in the scatter plot
leading to a $R^2$ of 0.53 with the LLR model and 0.45 with the LR model. The lower performance in the
year 2017 may be explained by an earlier drying conditions (June) compared with the previous years
with a dry season between August and September. However, both models seem able to predict RPoD
out of the calibration period.





### 3.3. Application of regional models

#### 3.3.1. Modeling of intermittencies severity between 2012 and 2016

Both models have been applied using the 2011-2017 dataset. Figure 11 displays the maximum number of consecutive days ($D_{RPoD>20\%}$) with RPoD higher than 20% simulated by both LLR and LR models. The most affected regions are located in the south-east of France and in sedimentary plains which are consistent with the spatial pattern obtained from the ONDE observations (Fig. 5). The most impacted year followed the same hierarchy: the year 2012 is the most critical year with 30% of France displaying $D_{RPoD>20\%}$ higher than 60 days followed by the year 2015 (20% of France with $D_{RPoD>20\%}$ > 60 days) and 2016 (15% of France with $D_{RPoD>20\%}$ > 60 days) (Fig. 11). The years 2013 and 2014 are weakly affected with 5% and 6% of the France with $D_{RPoD>20\%}$ higher than 60 days, respectively.

The LR model tends to simulate shorter periods of drying, particularly in HER2-HR combinations located in the South-East France in 2013 and 2014 (Fig. 11). However, there is an overall agreement between RPoD simulated by both models in terms of spatial and temporal extension of dry streams.

#### 3.3.2. Reconstitution of historical regional probability of drying

The trend temporal patterns of RPoD predicted by the two models, considering the 1989-2017 dataset, look similar between 1989 and 2016 and the simulated RPoD fit well to RPoD$_{ONDE}$ (Fig. 12). The results confirm the ability to reproduce the current conditions with both models, as discussed previously (Tab. 2).

The proportion of drying is highly variable over the total simulation period, with alternating dry (1989 to 1991, 2003 to 2006, 2009 to 2012) and wet (1994 to 1995, 2000 to 2002; 2013 to 2014) phases. In spite of interannual variability, peaks of RPoD occur regularly between August and September, whether in dry years or wet years. This finding is consistent with the preeminence of rainfall fed river flow regime with low flows in summer, in France.



The highest values of RPoDs (above 35% over France) are observed in 1989, 1990, 1991, 2003 and
2005 (black curve, Fig. 12a and b). The RPoDs simulated during these dry years are out of the range
of the observed values over the calibration period (2012-2016). Estimations are thus uncertain.
However, the high values of RPoD are consistent with observations reported in previous studies (e.g.
Larue and Giret, 2004; Snelder et al., 2013; Caillouet et al., 2017). Conversely, the years less affected
by drying are simulated in 1994, 2001 and 2014 with an average RPoD below 15% throughout the
year (black curve, Figs. 12a and b).
Results obtained with the LLR model are more contrasted in terms of extreme values than those
obtained with the LR model (Fig. 12b).

## 4. Discussion


*ONDE network complementarity with conventional flow monitoring network*
The analysis of the ONDE observations shows that the proportion of rivers undergoing drying is
significantly higher (35%) than that observe with the conventional monitoring (HYDRO database, 8%).
This proportion although related to a short period of records 2012 and 2016 is consistent with the
percentage of 39% of river segments classified as intermittent by Snelder at al. (2013). This analysis
confirms the under-representation of IRES in the French HYDRO database. Without any information
on headwater, Snelder et al. (2013) were unable to predict IRES in eastern France (see Fig. 9, pp.
2694). The high density of ONDE sites makes it possible to improve the dryings detection and lead to
better capture spatial distribution of IRES located at the upstream the hydrographic network. The
ONDE network encompasses various hydrological conditions which provides a more accurate
assessment of inter-annual variability, differentiating between dry years (2012, 2015 and 2016) and
wet years (2013, 2014) with clearly few drying occurrences.
The validation of the LR and LLR models against the POC database demonstrates also the
representativeness of the ONDE network. Thanks to the qualitative information provided and to





models such as statistical models developed here, it is now possible to capture drying event at the
regional scale.
The ONDE sites are located on small headwater streams which can be very reactive to external
disturbances (rainfall deficit, change in air temperature, increase in water withdrawals, etc.) and by
nature are more likely to be IRES. The gauging stations available in the HYDRO database are located
on larger streams and their hydrologic response to changes in external factors (environmental or
human) is slower and drying occurred with greater inertia under temperate climate. Their uneven
distribution across France does not allow to accurately characterize the inter-annual variability of
drying development.
ONDE network provides very complementary information to conventional flow monitoring, leading
to a better understanding of the processes of drying in upstream catchments.
*Dependency on spatial gauging networks density*
The performance obtained with the LR and LLR models is slightly better with the 2011-2017 dataset
(mean NSE ~ 0.75) than those obtained with the 1989-2017 dataset (mean NSE > 0.65), whose
network is less dense. HER2-HR combinations are the most degraded where the number of
monitoring stations is the most decreased between the two datasets. The accuracy of the predictions
is dependent on the number of gauging stations, ONDE sites and piezometers available to calibrate
the regressions. Highest NSEs are obtained in western sedimentary plains and southeastern of France
where a significant number of streams have dryings regardless of years (Fig. 5). The dominant river
flow regime in these regions is mainly influenced by precipitation and the lowest water levels are
reached in August and September, which corresponds to the monitoring period of the ONDE
database. They benefit from a dense monitoring network (gauging stations, ONDE sites,
piezometers), which allows a better representation of the hydrological functioning of streams
located within the same HER2. Conversely, performance was poor in mountainous areas such as in
the Alps or the Massif Central (NSE < 0.4) where river flow regimes are diversified combining rainfall



and snowmelt influences. By construction, the area of HER2-HR combination in mountains is
reduced, which leads to a limited number of monitoring stations, certainly not sufficient to fit the
models. Moreover, the observation period for ONDE sites was limited between May and September
and dryings can be missed, particularly for streams influenced by snow or glaciers melting with low
flows in winter.
We have chosen to average the non-exceedance frequencies of flows and groundwater levels in
order to increase the monitoring network. If models have been calibrated using only gauging
stations, performance will have been globally similar, or slightly better, in some HER2-HR
combinations (Fig. 13). There is thus no conclusion on a possible bias due to the use of piezometers.
This is certainly due to the dominant proportion of the gauging stations compared to the
piezometers. Indeed, in the 2011-2017 dataset, the proportion of gauging stations is greater than
75% for more than 70% of HER2-HR combinations whereas the proportion of piezometers exceeds
70% in only 5% of HER2-HR combinations. Groundwater level data thus have small weight in
regressions for this dataset. However, in the 1989-2017 dataset, the proportion of piezometer is
greater than 70% in more than 30% of HER2-HR combinations. The presence of piezometers
increases the density of the monitoring network in HER2-HR combinations with few available gauging
stations. Thanks to groundwater level data, RPoD can be predicted on more HER2-HR combinations.
*Interest in reconstructing the dynamic regional probability of drying*
Spatio-temporal simulation of the probability of drying is crucial for advancing our understanding of
IRES ecology and management. Some aquatic species can persist in dry reach for a few days, weeks
or months, while some are highly sensitive to desiccation (Datry, 2012; Storey and Quinn, 2013;
Stubbington and Datry, 2013). Estimating the total duration of the drying at the reach scale is
therefore needed to understand biological patterns in river networks. In France, based on the 2011-
2017 dataset, both models suggest highest values of RPoD along the Mediterranean coast
($D_{RPoD>20\%}$ > 100 days each year). Rivers in this region are subject to a predominantly pluvial regime





(Class 7; Sauquet et al., 2008), i.e. hot and dry summers follow by intense rainfall events in autumn,
leading to high flows in November (Skoulikidis et al., 2017b). The catchments in this region are small
and particularly reactive to environmental changes, making them highly sensitive to flow
intermittence. Rivers located in sedimentary plain in western France are also very impacted by flow
intermittence. The regime is also influenced by precipitation and the basins are subject of intense
agriculture with important water withdrawals during summer. Abstractions greatly reduce the water
availability in rivers and in aquifers which are no longer able to support the low water levels and lead
to increased flow intermittence. The responses of biological communities to artificial flow
intermittence is still poorly understood compared to natural IRES (Datry et al., 2014b, Skoulikidis et
al., 2017a).
*Validity of historical regional probability of drying during severe low-flow period*
The second application aimed at reconstructing historical RPoD over the period 1989-2016. Both
models suggest highest values of mean RPoD (> 35%) in 1989-1991, 2003 and 2005. During these dry
years, predicted values of RPoD results from extrapolation but are consistent with published studies
(Mérillon and Chaperon, 1990, Moreau, 2004). For example, Mérillon (1992) estimated that for the
whole of France, 11 000 km of rivers were dried at the end of summers of 1989 and 1990. Caillouet
et al (2016) found that the low-flow event observed in 1989-1990 was particularly severe in terms of
duration and affected 95% France territory. Snelder et al. (2013) showed from 628 gauging stations
that the years 1989-1991, 2003 and 2005 had witnessed particularly high values of duration and
frequency of drying. They found that regions with the highest probability of drying were located
along the Mediterranean and Atlantic coasts, which is consistent with ONDE observations and with
our results.
Both models suggest the same sequence of dry and wet years. However the application of the LLR
model lead to less contrasted RPoD than LR model (Fig. 12).




To illustrate these differences, the RPoD has been simulated by both models with a fictive extreme F
of 1% (Fig. 14). The $RPoD_{LLR}$ is significantly higher and exceeds 80% in 30% of the France territory
against only 5% of the territory with the $RPoD_{LR}$. On the other hand, models simulate low RPoD in
HER2-HR combinations where the $RPoD_{ONDE}$ is very low between 2012-2016, even when F was 1%
because this situation never occurred during the calibration period (Fig. 14). The logistic function of
the LR model takes an S-shape which induced a crushing of extreme values around the maximal
values observed during the calibration period (2012-2016). The truncated logarithmic function of the
LLR model is not bounded and RPoD can reach 100% during extreme low flow events by
extrapolation. ONDE network has not witnessed a severe low-flow period as in 1990. In that sense, it
is not currently possible to differentiate between the two models similar performances over the
same calibration period. Refining extrapolated values require additional information on headwater
collected during more severe droughts than those observed during the last five years and then gives
support to the pursuit of the ONDE network.
## 5. Conclusion
This paper investigates the spatial and temporal dynamics of the regional probability of drying (RPoD)
of headwater streams by taking benefit from qualitative and discontinuous data provided by the
ONDE network. Two models based on linear or logistic regressions have been used to reconstruct the
temporal dynamics of RPoD. They are based on a strong relationship between the non-exceedance
frequencies of discharges and groundwater levels as a function of the proportion of drying observed
at ONDE sites per HER2-HR combination. LLR and LR models show similar performance and perform
well between 2011 and 2017. The accuracy of predictions is dependent on the number of gauging
stations, ONDE sites and piezometers available to calibrate the regressions. Regions with the highest
performance are located in sedimentary plains, where the monitoring network is dense and where
the RPoD is the highest. Conversely, the worst performances are obtained in the mountainous
regions. Finally, both models have been used to reconstitute historical RPoD between 1989 and 2016





and suggest highest values of mean RPoD (> 35%) in 1989-1991, 2003 and 2005. This is consistent
with other published studies but the high density of ONDE sites makes it possible to improve the
dryings detection and lead to better capture the spatial distribution of IRES located at the upstream
the hydrographic network.
The next step will be to use this regional approach to simulate the RPoD in future periods by taking
into account effects of climate change through predicted discharge and groundwater level data. This
would allow to quantify the evolution of the probability of drying between the current period and
the different climate projections provided by the latest IPCC Report (IPCC 2014a, 2014b) report and
would assist decision makers in defining protocols for restoring flows with appropriate measures to
preserve aquatic ecosystems (Woelfle-Erskine, 2017).
Secondly, further work is needed to develop an approach capable of reconstructing the drying
dynamics locally by differentiating each stream. This will allow decision makers to take more
appropriate measures to limit locally and more effectively the IRES development and thus preserve
the biodiversity of these environments. Our approach remains spatially valid to estimate RPoDs at
the scale of HER2-HR combinations and does not allow characterizing the variability of the probability
of drying occurrence between nearby streams within these regions. However, previous studies have
shown that this could be very complex because diverse flow states (lentic, lotic, flowing) could
develop in the same segment of river separated by small distances and create highly fragmented
hydrographic networks (Datry et al., 2016a). It is necessary to take into account many drivers
involved in the flow intermittence development such as: rainfall, potential evapotranspiration, air
temperature, exchanges between open channels and groundwater systems, riparian vegetation,
water abstraction, river bed topography, etc. These variables are rarely available at fine scales at a
country scale. From a methodological point of view, statistical tools such as neural networks
(Breiman, 2001) have shown good ability to assess both the occurrence and extent of perennial and



temporary segments (González-Ferreras and Barquín, 2017) and could be investigated as an
alternative method to reconstruct locally the temporal variability of drying.

## 6.  Acknowledgment

The research project was partly funded by the French Agency for Biodiversity (AFB, formerly
ONEMA). This study is based upon works from COST Action CA15113 (SMIRES, Science and
Management of Intermittent Rivers and Ephemeral Streams, www.smires.eu), supported by COST
(European Cooperation in Science and Technology).





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




|  | Stations with at least one drying event | Stations with drying > 50% | Frequency of discharge < 1 l/s |
|---|---|---|---|
| **2012** | 79 | 19 | 32.7 |
| **2013** | 47 | 14 | 37.9 |
| **2014** | 54 | 15 | 32.9 |
| **2015** | 76 | 21 | 31.1 |
| **2016** | 71 | 19 | 28.6 |

**Table 1.** Annual statistics on flow intermittence calculated on HYDRO gauging stations between the
1$^{st}$ May and the 30$^{th}$ September





| | | 2011-2017 dataset | | | | | 1989-2017 dataset | | | | | |
|---|---|---|---|---|---|---|---|---|---|---|---|---|
| | | 2012 | 2013 | 2014 | 2015 | 2016 | **2017** | 2012 | 2013 | 2014 | 2015 | 2016 | **2017** |
| **LLR model** | May | 0.2 | 0.0 | 0.5 | 0.5 | 0.6 | **0.4** | 0.2 | 0.0 | 0.3 | 0.0 | 0.7 | **0.2** |
| | June | 0.6 | 0.3 | 0.8 | 0.5 | 0.8 | **0.5** | 0.6 | 0.3 | 0.5 | 0.3 | 0.8 | **0.5** |
| | July | 0.7 | 0.5 | 0.6 | 0.6 | 0.8 | | 0.7 | 0.5 | 0.5 | 0.4 | 0.8 | |
| | August | 0.8 | 0.6 | 0.7 | 0.7 | 0.8 | | 0.7 | 0.5 | 0.5 | 0.5 | 0.8 | |
| | Sept. | 0.7 | 0.8 | 0.6 | 0.6 | 0.7 | | 0.6 | 0.7 | 0.5 | 0.5 | 0.6 | |
| | May - Sept | 0.8 | 0.8 | 0.7 | 0.7 | 0.8 | **0.5** | 0.8 | 0.7 | 0.5 | 0.6 | 0.8 | **0.5** |
| **LR model** | May | 0.2 | 0.0 | 0.5 | 0.1 | 0.6 | **0.3** | 0.3 | 0.0 | 0.3 | 0.0 | 0.7 | **0.2** |
| | June | 0.6 | 0.5 | 0.8 | 0.5 | 0.8 | **0.4** | 0.6 | 0.4 | 0.5 | 0.3 | 0.7 | **0.4** |
| | July | 0.7 | 0.6 | 0.5 | 0.6 | 0.8 | | 0.7 | 0.4 | 0.5 | 0.4 | 0.8 | |
| | August | 0.7 | 0.6 | 0.7 | 0.6 | 0.7 | | 0.6 | 0.4 | 0.5 | 0.4 | 0.7 | |
| | Sept. | 0.6 | 0.8 | 0.6 | 0.7 | 0.7 | | 0.5 | 0.6 | 0.4 | 0.5 | 0.6 | |
| | May - Sept | 0.8 | 0.8 | 0.7 | 0.7 | 0.8 | **0.4** | 0.8 | 0.7 | 0.5 | 0.6 | 0.8 | **0.4** |

**Table 2.** NSE criteria obtained between 2012 and 2017 with the LLR and LR models calibrated over
663                                              the period 2012-2016.






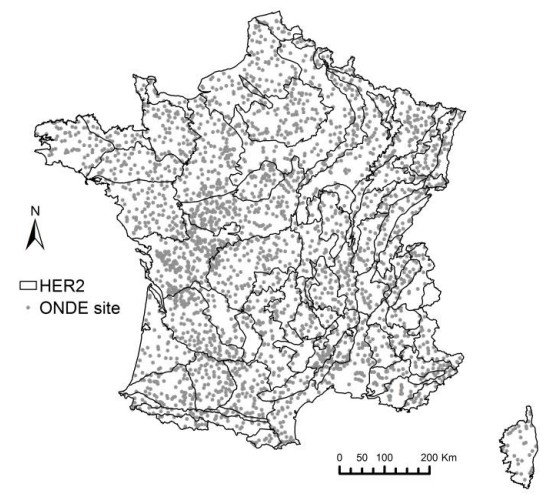


666        **Figure 1.** Location of the 3 300 ONDE sites and partition into HER2.


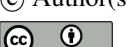



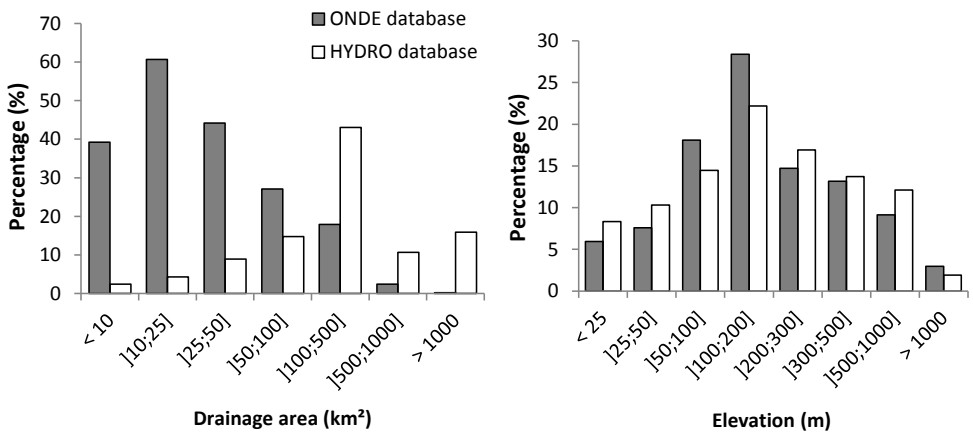

**Figure 2.** Distribution of the 3 300 ONDE sites and of the 1 600 gauging stations available in the HYDRO database against: (a) drainage area and (b) elevation.





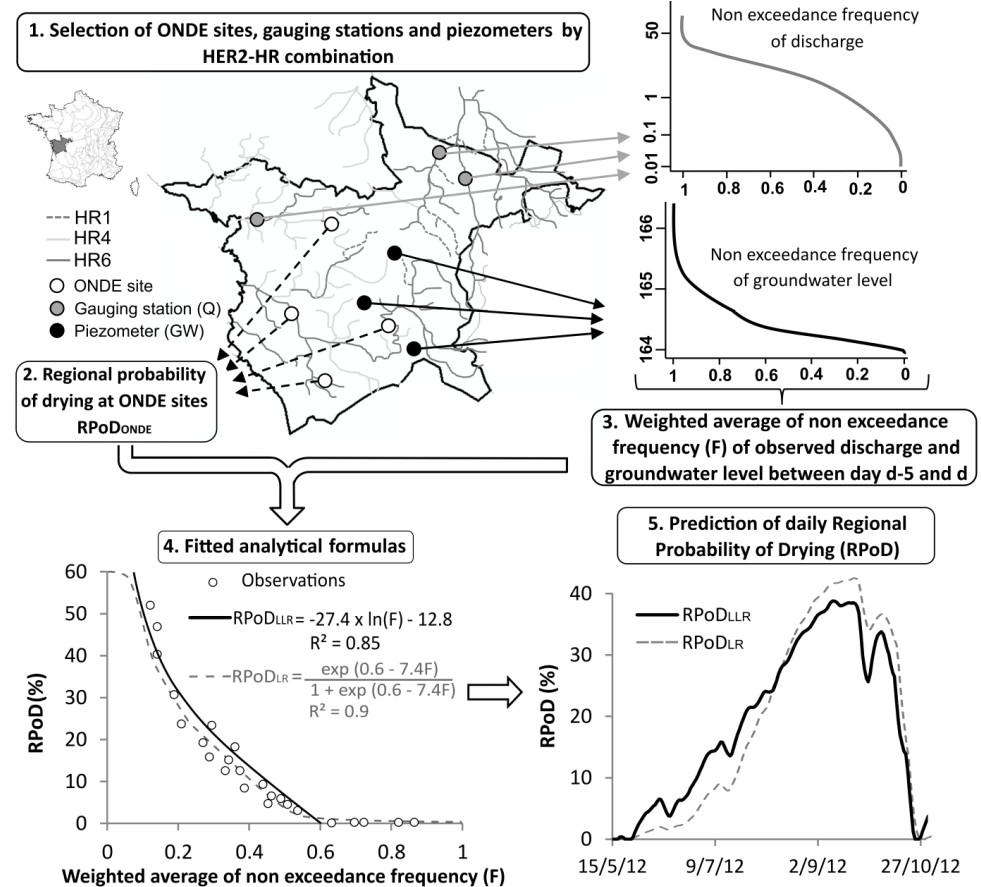

**Figure 3.** Strategy of parametric modeling (step 1-4) developed to predict (step 5) the regional
probability of drying (RPoD) by HER2-HR combination in France.



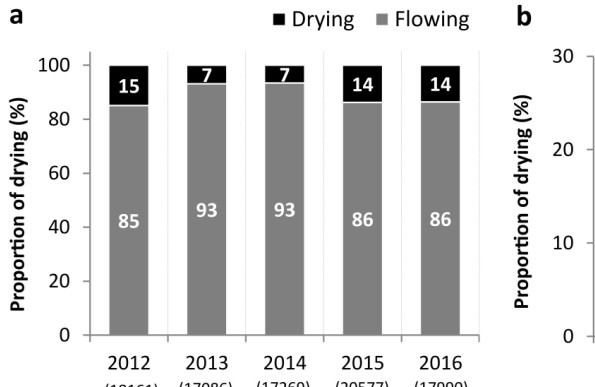
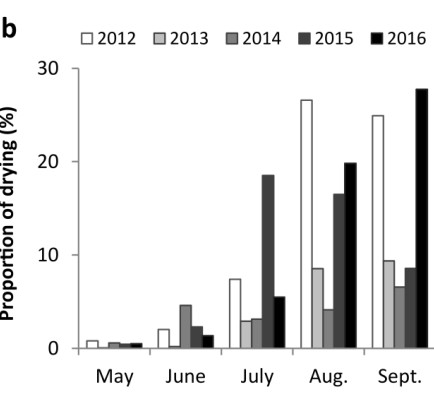


**Figure 4.** (a) Distribution of yearly proportion of drying observed with the ONDE network with the
total yearly number of ONDE observations written in brackets and (b) distribution of proportions of
679                                        drying per year and per month.





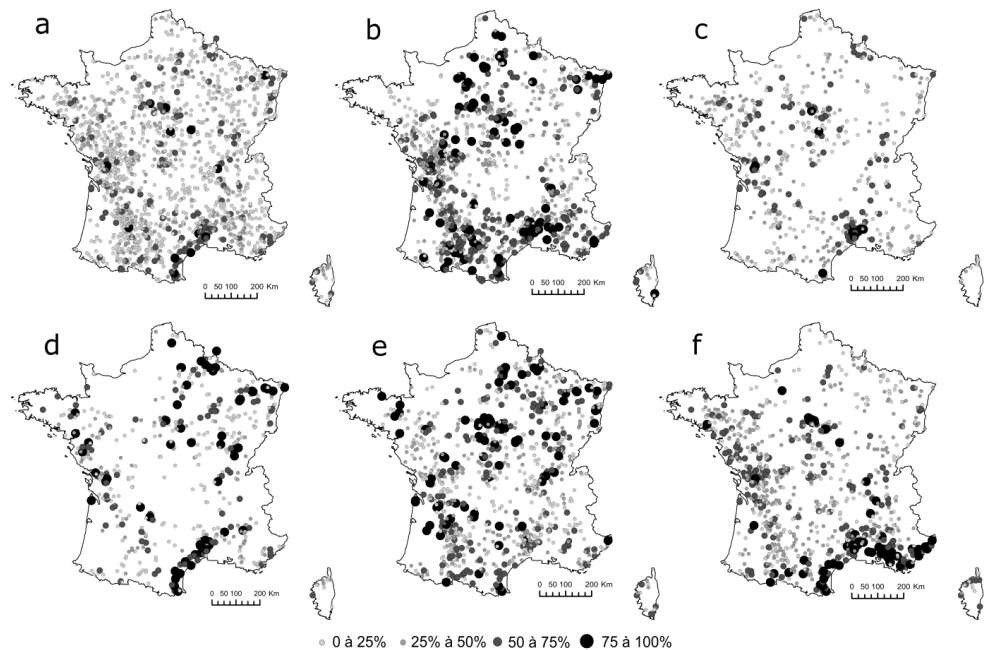


**Figure 5.** Distribution of the percentages of drying observed at ONDE sites for the years: (a) 2012-2016, (b) 2012, (c) 2013, (d) 2014, (e) 2015 and (f) 2016.






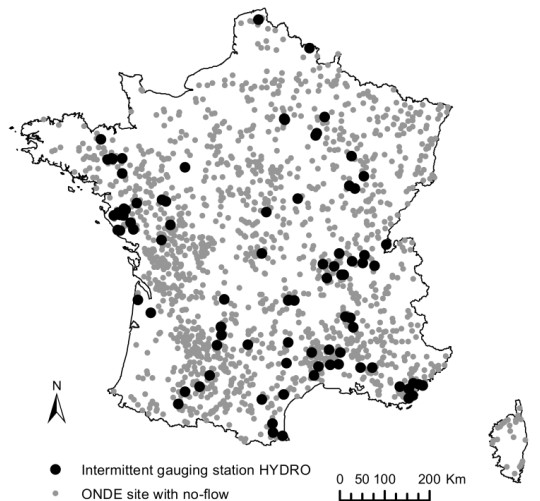


**Figure 6.** Map of ONDE sites and HYDRO gauging stations having at least one drying.




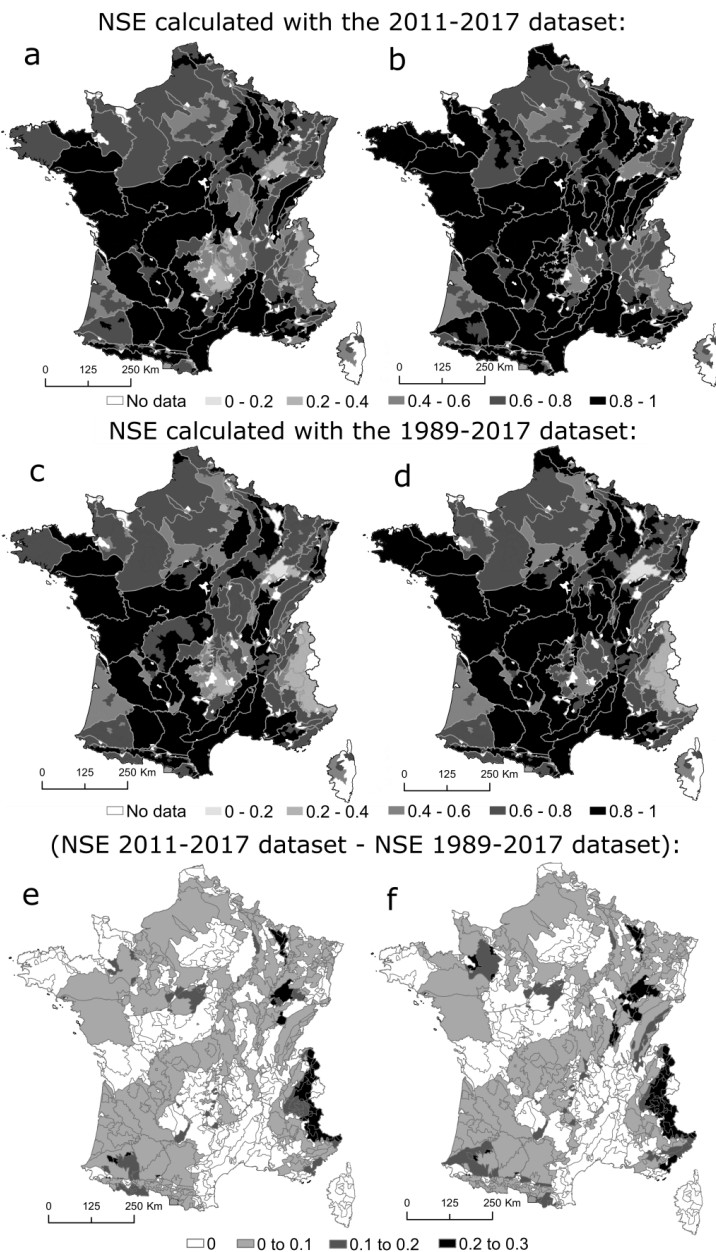


**Figure 7.** Map of Nash-Sutcliffe criteria (NSE) obtained for each HER2-HR combination between 2012
and 2016 with the 2011-2017 and 1989-2017 datasets according to: (a) and (c) a log-linear regression
(LLR) model; (b) and (d) a logistic regression (LR) model. NSE differences between the 2011-2017
dataset and the 1989-2017 dataset are represented for: (e) LLR model and (f) LR model.

693





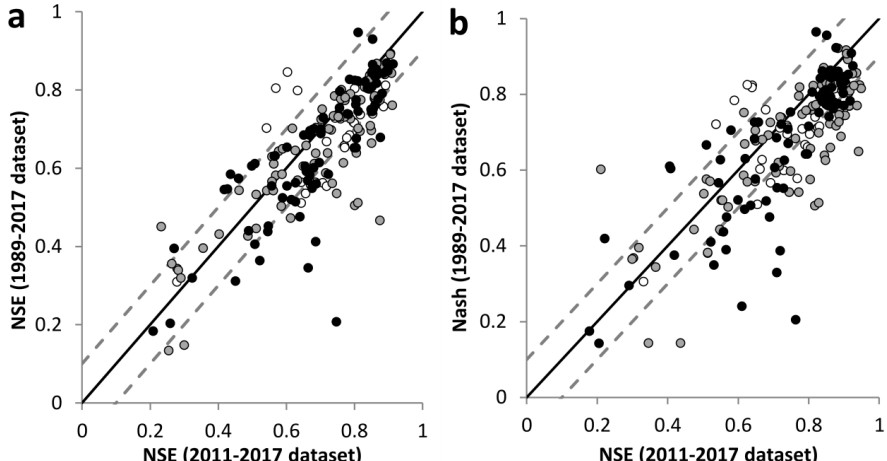

694

**Figure 8.** Nash-Sutcliffe criteria (NSE) calculated for each HER2-HR combination between 2012 and
2016 with the 1989-2017 dataset as a function of NSE calculated with 2011-2017 dataset with
respectively: (a) the LLR model and (b) the LR model. The color of dots represents the proportion of
gauging station and piezometers lost between the 2011-2017 database and the 1989-2017 database:
losses < 50% (white); losses between 50% and 75% (grey); losses > 75% (black).






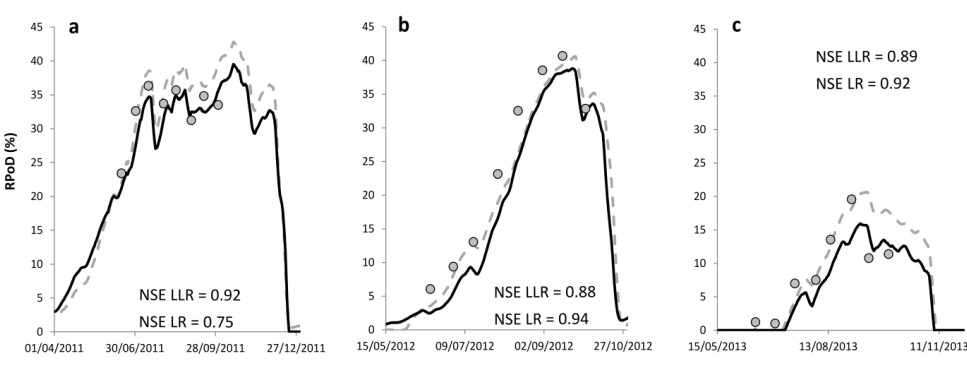

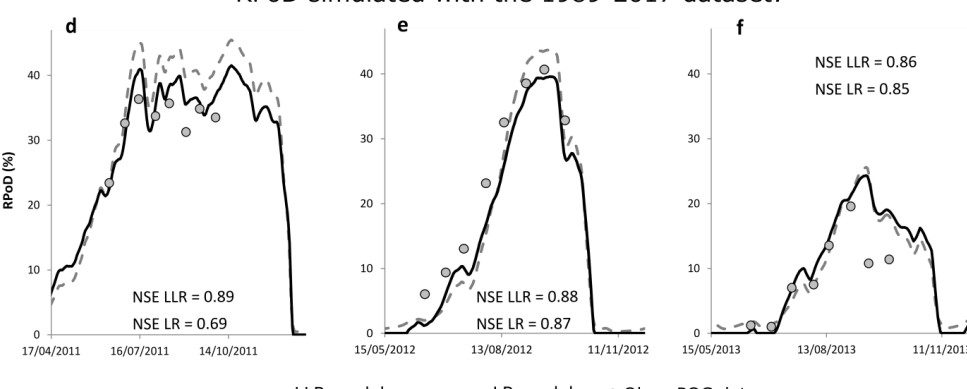


**Figure 9.** Comparison between observed proportion of drying RPoD$_{POC}$ and RPoD predicted by the LLR
703        and LR models with the 2011-2017 dataset in: (a) 2011, (b) 2012 (c) 2013 and with the 1989-2017
704                        dataset in: (d) 2011, (e) 2012 (f) 2013.






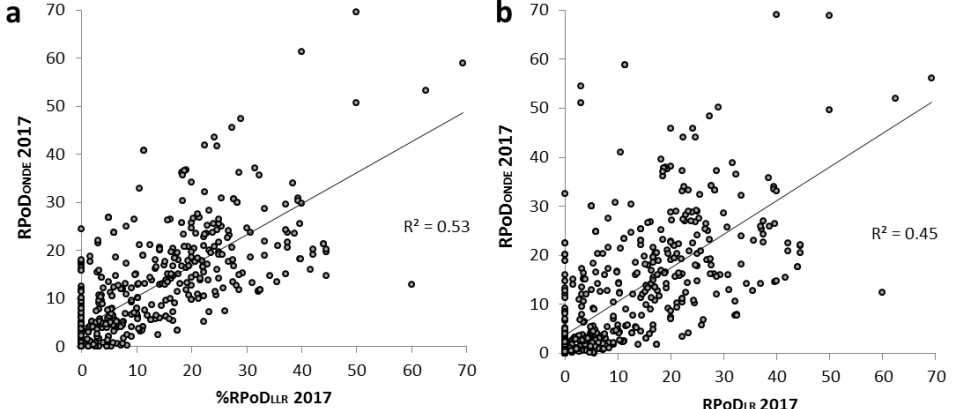


**Figure 10.** Scatter plot of the predicted RPoD (x axis) and drying observed at ONDE sites (y axis) in
2017 simulated with the 2011-2017 dataset by: (a) the LLR model and (b) the LR model.




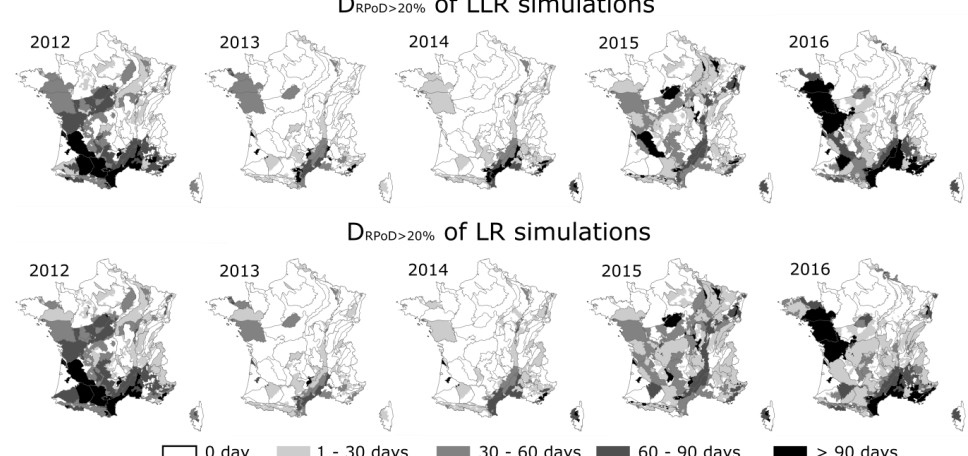


**Figure 11.** Maximum duration of consecutive days with RPoD higher than 20% simulated with LLR
and LR model.





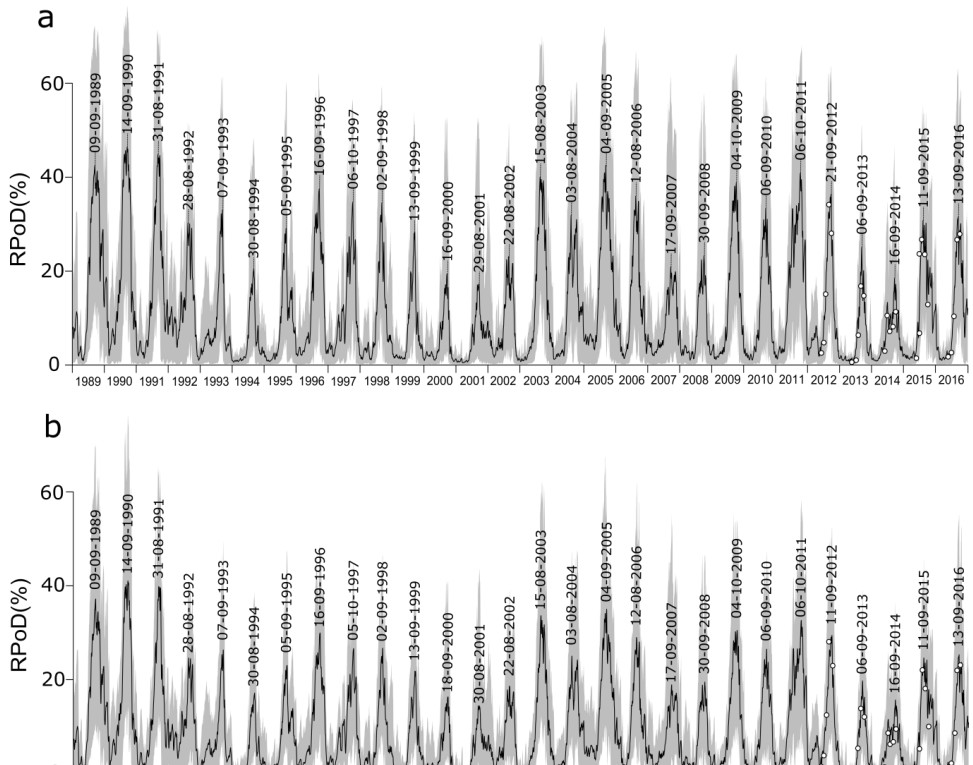


**Figure 12.** RPoD simulated between 1989 and 2016 with: (a) the LR model and (b) the LLR model. The
grey area represents the RPoD between the $90^{th}$ percentile and the $10^{th}$ percentile simulated on
HER2-HR combination, the black curve represents the average RPoD simulated by HER2-HR
combination and white dots represent the mean $RPoD_{ONDE}$ for each observation dates.






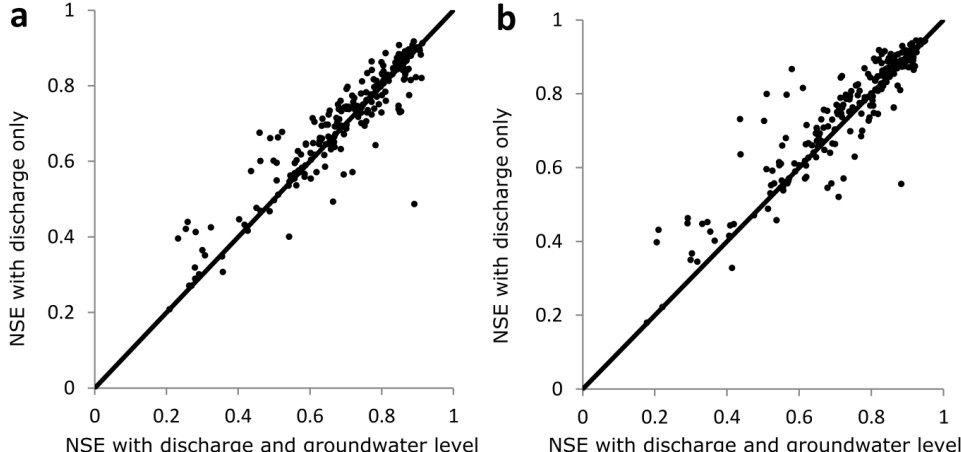


**Figure 13.** Comparison of NSE obtained with regression including only discharge variable as a function of NSE obtained with including discharge and groundwater level variables in the 2011-2017 dataset with: (a) LLR model and (b) LR model.






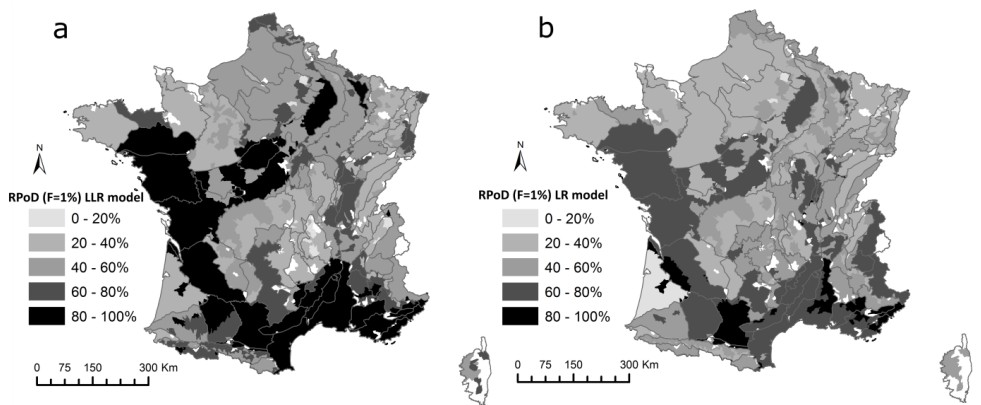


**Figure 14.** Regional probability of drying simulated with F = 1% predicted with: (a) the LLR model and (b) the LR model.









