# Peer review of "Extrapolating regional probability of drying of headwater streams"

_Hydrology and Earth System Sciences, 2017_

## Referee Comment (RC1) · A.F. Van Loon (Referee) · 22 Dec 2017

**Review of BEAUFORT et al. "Extrapolating regional probability of drying of headwater streams using discrete observations and gauging networks"**

**By Anne Van Loon**

**General comments:**

I would like to congratulate the authors with this interesting paper. In the paper, they use a number of databases (official networks and citizen science data) to predict regional drying of headwaters in France, which gives interesting information on spatial and temporal variability of drying. The data, approach and results are robust. I do have a few fundamental and technical questions (see below), but I hope these can be solved easily by the authors.

Firstly, the authors need to explain why a regional assessment of headwater drying is needed. What is the benefit of Figure 11 over Figure 5? The patterns of drying are the same, so Figure 5 would be sufficient to indicate hotspots of drying within France and temporal variability in drying. In the discussion, the authors point out that for accurate IRES management estimation of "drying at the reach scale is needed" (p.18 l.427) and in the conclusion they mention that the approach does not allow for characterisation of drying in "nearby streams within the regions" (p.21 l.495). So if local scale information is so important and this method cannot be used to extrapolate between streams in one region, then why do we need the regional scale? Why go to coarser resolutions if you have detailed observation data at least for some rivers? In this way you lose spatial information without gaining anything in return.

Secondly, the paper is focused on France. This in itself is not a problem, since the methodology and results are interesting and useful beyond France, but the author fail to put their findings in a broader perspective in the discussion. Literature on IRES research from outside France should be discussed and the authors should clarify what is new and interesting about this work from an international perspective. On p.19 l.443-452, the authors mention how their results are consistent with previous studies, which is great, but they should additionally point out what their study adds. If this is not done, the study would be better placed in a Journal like Journal of Hydrology – Regional Studies.

Thirdly, I would like the authors to help the reader more in understanding the methodology. Figure 3 is helpful, but in the manuscript it is not always clear which data was used for what. Especially when explaining the equations on page 9 and 10, the authors could be clearer on which dataset was used, which time period. Also in the Results section it should be clarified when they are referring to calibration results, validation with POD data, or validation with the year 2017. For example, the first paragraph of Section 3.2.3 is quite confusing, because it discusses the performance of the models in the calibration period, which was already discussed in Section 3.2.1. Table 2 should be explained better; how is it different or similar to the information presented in Figures 7&8? Also, in the first paragraph of Section 3.3.2 the authors state that "the simulated RPoD fit well to RPoDONDE" (l.349), but wasn't that already discussed in Section 3.2.1 (Figure 7&8)?

Fourthly, it is unclear whether natural and/or human-influenced sites are selected in this study. In Section 2.4, the authors mention that the "observed discharges were not or only slightly altered by human actions" (p.7 l.164), but they do not specify whether the other datasets, i.e. groundwater

levels, ONDE and POD observations, are near-natural too. This is important, as the authors mention in the discussion, "the basins are subject of intense agriculture with important water withdrawals during summer. Abstractions greatly reduce the water availability in rivers and in aquifers which are no longer able to support the low water levels and lead to increased flow intermittence. The responses of biological communities to artificial flow intermittence is still poorly understood compared to natural IRES." (p.19 l.435-439) If near-natural and human-influenced data are mixed in the predictions, it will be very difficult to understand the reasons for the regional patterns in drying and the statements about the highest drying occurring in sedimentary plains due to the low elevation gradient and dependence on rainfall might be flawed.

And finally, it is unclear why two statistical models are used throughout the paper. If they are equally suitable from a theoretical perspective, two (or more) models could be used for testing, but then the best model should be used to simulate the final results.

**Specific comments:**

The regional probability of drying needs to be explained. In Section 2.6 the authors only mention that RPoD is calculated, but they never explain how this variable is calculated exactly.

The weighted average of the non-exceedance frequencies (F) needs to be explained better. According to the Discussion section discharge and groundwater levels are combined (l.411-412), but this is not explained clearly enough in the Methods section (l.202-203). How are these non-exceedance frequencies of groundwater and discharge averaged since they have such different shapes and ranges (see Figure 3). And what do the authors mean with "with respect to the relative proportions of gauging stations and piezometers" (l.203-204)?

The authors conclude that "both models seem able to predict RPoD out of the calibration period" (l. 330-331), but do a NSE of 0.4 and 0.5 warrant such a statement?

A significant part of the Conclusion section discusses future work. Is that relevant for this manuscript? I would suggest leaving those paragraphs out as they distract from the main message of this paper.

**Textual comments:**

- l.20-21: make the time periods consistent. The abstract mentions 2011-2017 and 1989-2017, whereas the introduction mentions 2012-2016 and 1989-2016 (l.97-98).

- l.155: RPoDpoc not explained yet. The concept of RPoD needs to be explained first, before you can introduce different versions.

- l.194: change "we merged 20 of the 280 regions with a neighboring one located in the same HER1" to "we merged the NER2-HR combination with a neighboring one located in the same HER1. This was done for 20 of the 280 regions."

- l.307: Fig. 10 > Fig. 9

- l.317: "both models use with the 1989-2017" > what was meant here?

- l.322: change "whatever the dataset" to "independent of dataset"

- l.372-373: this is an unclear sentence. What "information on headwaters" was missing? And why were they unable to predict?

- l.375: what do you mean with "better capture spatial distribution of IRES located at the upstream the hydrographic network"?

- l.460: reformulate "a crushing of extreme values"

- Fig. 8: Nash > NSE

- Fig. 12: explain dates in figure

- Please check the English of the manuscript, for example the consistency between verb and subject, e.g. l.248 "an increase", l.250 "reaches" & "occurs", l.369 "observed", l.465 "requires"

---

## Referee Comment (RC2) · C. Sefton (Referee) · 11 Feb 2018

This paper makes a valuable contribution to the study of the hydrology of intermittent rivers and ephemeral streams. In particular, it is the first time observations of river flow and groundwater level have been used in combination with observations of hydrological state in a regionalisation approach, marking a step forward in the modelling and mapping of intermittency at national scale.

The ONDE dataset is unique in the literature, notably the large number of sites, the coverage of headwater streams and the national extent, but has limitations in the summer-only timing and small number of observations. The merging of the "no visible flow"

status with the "dried out" status means that a key benefit of the dataset is not utilised, as the two-status classification of flowing and drying that remains is no advantage over that available from gauged river flow data. Discussion of the network would benefit from broader contextual comment on the contribution and application of these data.

The paper is well written and referenced and is recommended for acceptance with minor revisions.

Please also note the supplement to this comment:
https://www.hydrol-earth-syst-sci-discuss.net/hess-2017-630/hess-2017-630-RC2-supplement.pdf

**Supplement:**

*Interactive comment* on "Extrapolating regional probability of drying of headwater streams using discrete observations and gauging networks" by Aurélien Beaufort et al.

**Catherine Sefton (Referee), 11 February 2018**

**General comments**

This paper makes a valuable contribution to the study of the hydrology of intermittent rivers and ephemeral streams. In particular, it is the first time observations of river flow and groundwater level have been used in combination with observations of hydrological state in a regionalisation approach, marking a step forward in the modelling and mapping of intermittency at national scale.

The ONDE dataset is unique in the literature, notably the large number of sites, the coverage of headwater streams and the national extent, but has limitations in the summer-only timing and small number of observations. The merging of the "no visible flow" status with the "dried out" status means that a key benefit of the dataset is not utilised, as the two-status classification of flowing and drying that remains is no advantage over that available from gauged river flow data. Discussion of the network would benefit from broader contextual comment on the contribution and application of these data.

The paper is well written and referenced and is recommended for final publication with minor revisions.

**Specific comments**

Given the small number of observations at each site, the claim that the ONDE dataset offers more accurate assessment of inter-annual variability than the gauging station network (L376-377) needs further justification. Conversely, the claim that the dataset makes it possible to capture drying events at the regional scale (L381-382) would benefit from stressing the monitoring of both upstream and downstream drying – uniquely each with national extent – in your approach.

The presentation of summer-only status data as "% drying" needs qualification, as it suggests assumptions about the status in the rest of the year. In particular, clarification would be helpful in line 242, when the context implies it means the number of sites with at least one drying each year (as in line 241), rather than the % of all observations at all sites (as in Fig 4).

The technique of constructing mean non-exceedance frequency from river flows and groundwater level is attractive and robust. However, its limitation in this regional approach of failing to capture the effect of local rainfall should be commented upon, especially given the dominance of rainfall-driven intermittency stated in section 3.3.2.

The frequency of drying from gauging station data needs to be defined (line 272). Context suggests it is flow permanence (dry days or dry months per time period), but frequency in intermittent rivers and ephemeral streams can also mean dry spells per time period, and it also needs to be clear and justified whether it's calculated from daily means or monthly means.

In section 3.2.1., the difference in performance between the two explanatory hydrological datasets is attributed to the difference in the number of gauging stations and piezometers. The pattern in Figure 8

is not as clear as the text suggests, and it would be good to comment also on the assumption of stationarity and how it might vary between HER2-HR combinations. Similarly, historical reconstructions make assumptions about stationarity that need to be acknowledged.

The conclusion is a good summary of the results but would benefit from contextual comment, both with respect to the stated objective of this paper and more broadly on the contribution being made to the field.

**Text corrections**

| | |
|---|---|
| General | inconsistent hyphenation and capitalisation of "hydroecoregions" |
| L8 | "have frequently flows intermittence" -> "and frequently have flow intermittence" |
| L22 | "The two regressions models" -> "The two regression models" |
| L43 | "are referred as" -> "are referred to as" |
| L48 | "they represents" -> "they represent" |
| L51 | "severely underestimate" -> "severe underestimation of" |
| L73 | "citizen science has proved" -> "citizen science is proven to" |
| L79 | "in few catchments" -> "in a few catchments" |
| L88-89 | "and to characterise their information contribution in comparison with" -> "in order to characterise the information that they contribute in comparison with" |
| L141 | "more likely reason" -> "most likely reason, therefore" |
| L165 | "composed by" -> "composed of" |
| L186 | "groundwater level were" -> "groundwater level was" |
| L230 | "have been calibrating" -> "have been calibrated" |
| L235 | "be noted that models predictions" -> "be noted that model predictions" |
| L245 | "in more details -> "in more detail" |
| L247-8 | "but follow the same pattern with an increased…and reached…" -> "but follows the same pattern with an increase…and reaching…" |
| L250 | "August 4% and reach 7%...critical period occur" -> "August (4%), reaching 7%...critical period occurs" |
| L252 | "until reaching…situation is gradually deteriorated" -> "until it reaches…situation gradually deteriorates" |
| L264 | "HYDRO dataset" -> "The HYDRO dataset" |
| L317 | "when both models use with the" -> "when both models use" |

| L346 | "temporal and spatial extension" -> "spatial and temporal extent" |
|---|---|
| L373 | "headwater" -> "headwaters" |
| L374/482 | "dryings detection" -> "detection of drying" |
| L375/482 | "capture" -> "capturing of the" |
| L375/483 | "upstream" -> "upstream extent of" |
| L434 | "in sedimentary" -> "in the sedimentary" |
| L444 | "results" -> "result" |
| L448 | "France territory" -> "of France" |
| L454 | "than LR model" -> "than the LR model" |
| L455 | "To illustrate these differences…with a fictive extreme F" -> "In a sensitivity test to illustrate these differences …with an extreme F of" |
| L456 | "France territory" -> "study area" |
| L457 | "of the territory" -> "of the area" |
| L463 | "ONDE network…it is not currently possible to differentiate between the two models similar performances" -> "Since the ONDE network monitoring period does not include a period with flows as low as were seen in 1990, it is not currently possible to differentiate between the two models' performances" |
| L479 | "reconstitute" -> "reconstruct" |
| L486 | "would allow to quantify the" -> "would allow quantification of the" |
| L487 | ") report and" -> ") and" |
| L494 | "and" -> "but" |

**Figures**

| Figure 3, textbox 3 | The text is truncated |
|---|---|
| Figure 5 | The legend is in French |
| Figure 3, step 1 | It is unclear why HR1, 4 and 6 are shown as types of monitoring site, when section 2.1 has defined them as types of hydrological regime. |
| Figure 9 | Referenced as Figure 10 in line 307 |
| Figure 10 | This would benefit from additional plots for a year that has good NSE, as the text is comparing years as well as model performance. |
| Figure 12 | The caption does not say which explanatory dataset was used. |

---

## Author Comment (AC1) · 6 Mar 2018

**Reply to Anne Van Loon**

*I would like to congratulate the authors with this interesting paper. In the paper, they use a number of databases (official networks and citizen science data) to predict regional drying of headwaters in France, which gives interesting information on spatial and temporal variability of drying. The data, approach and results are robust. I do have a few fundamental and technical questions (see below), but I hope these can be solved easily by the authors.*

The authors would like to thank Anne Van Loon for her positive comments on the manuscript. Please find below the detailed answers to the general and specific comments.
* * *
**General comments**

*Firstly, the authors need to explain why a regional assessment of headwater drying is needed. What is the benefit of Figure 11 over Figure 5? The patterns of drying are the same, so Figure 5 would be sufficient to indicate hotspots of drying within France and temporal variability in drying.*

We modified lines 87-93 to better explain that one of our objective is a temporal extrapolation of the daily drying probability (eg Fig 11) in regions, based on discrete observations (~5/years, raw data in Fig 5).

Figure 11 shows the number of consecutive days with simulated RPoD > 20% to characterize the severity in both time and space of drying. Figure 11 is derived from the reconstructed continuous time series. Figure 5 results from ONDE observations, i.e. statistics are based on five inspections per year between May and September.

As discrete data, ONDE observations cannot provide any information on the persistence of dry conditions between two consecutive dates of observation. In summer, rewetting is possible after convective rainfall episodes or inversely short-lived events of drying may occur between two dates with "Flowing" state. The rewetting-drying events may have significant impacts on communities whose survival is conditioned by the duration/frequency of drying. The duration of drying is of importance for ecologists, as one key driver of the composition and persistence of aquatic species (e.g. Kelso and Entrekin, 2018). During recent decades, hydrologists and ecologists have been working on developing metrics to quantify alterations of the river flow regime and their consequences for the ecosystems (Poff and Ward, 1990; Richter et al., 1996; Snelder et al., 2009; De Girolamo et al., 2017). Most of the metrics (e.g. Poff et al., 1997; Olden and Poff, 2003; D'Ambrosio et al., 2017) are determined on the basis of continuous time series of daily discharge.

In that sense, the objective of this study is to provide information through probability of drying to a daily time step using discontinuous data from the ONDE network.
* * *
*In the discussion, the authors point out that for accurate IRES management estimation of "drying at the reach scale is needed" (p.18 l.427) and in the conclusion they mention that the approach does not allow for characterisation of drying in "nearby streams within the regions" (p.21 l.495). So if local scale information is so important and this method cannot be used to extrapolate between streams in one region, then why do we need the regional scale? Why go to coarser resolutions if you have detailed observation data at least for some rivers? In this way you lose spatial information without gaining anything in return.*

This work is the first step towards a more ambitious project: modelling the dynamic of daily flow-states at the ONDE sites, *i.e.* at the reach scale. The idea was to start by the coarsest spatial scale before developing tools adapted to the local scale. Thus, the effort was mainly put on the temporal aspects but at the regional scale. It has consisted in identifying the robust and significant indices that are related to flow intermittence at the regional scale. The indices allow extrapolating information from the discrete observation data and will be introduced in the set of potential explanatory variables or considered as proxies of the probability of dry conditions in the modelling framework for each ONDE site. Other local potential explanatory drivers such slope, riparian vegetation, presence of pools, rainfall, evapotranspiration, permeability, water abstraction, etc. that are scarcely available at the country scale, will be collected in the next steps.
* * *
*Secondly, the paper is focused on France. This in itself is not a problem, since the methodology and results are interesting and useful beyond France, but the author fail to put their findings in a broader perspective in the discussion. Literature on IRES research from outside France should be discussed and the authors should clarify what is new and interesting about this work from an international perspective. On p.19 l.443-452, the authors mention how their results are consistent with previous studies, which is great, but they should additionally point out what their study adds. If this is not done, the study would be better placed in a Journal like Journal of Hydrology – Regional Studies.*

The authors agree with this remark. We modified the text (lines 465-481 and lines 544-551) to better explain the international relevance of our results.

To our knowledge, no study has proposed to reconstruct daily flow states time series of headwater streams at the country scale as France (> 500 000 km²) using discrete observations in time and space. In the literature, studies at national scale remain focused on the detection and the mapping of IRES because these rivers are historically poorly investigated and their proportion in existing hydrographic networks remains inaccurate or misunderstood (Nadeau and Rains, 2007; Snelder et al., 2013). Recently, several studies proposed alternative methodologies in order to estimate metrics in ungauged IRES (Gallart et al., 2016) or to predict daily streamflow in river basin experiencing flow intermittence (De Girolamo et al., 2017) but remain applicable at local scale.

From a methodological point of view, our method relating discrete drying observation to continuous daily gauging data seems robust across the highly diverse (climate and topography) regions of France, and provides good predictions in an independent region excluded from the calibration process (PoC). These two results suggest a potential application of our approach in other countries.

Citizen science has proved to create opportunities to overcome the lack of hydrological data and lead to densify the flow state observation network (Turner and Richter, 2011; Buytaert et al., 2014). Note that the paper demonstrates the value of the ONDE network and thus promotes such a kind of network whose creation is less expensive than the installation of additional gauging stations, to survey flow intermittence.

\*\*\*\*\*\*\*\*\*\*\*\*\*\*\*\*\*\*\*\*\*\*\*\*\*\*\*\*\*\*\*\*\*\*\*\*\*\*\*\*\*\*\*\*\*\*\*\*\*\*\*\*\*\*\*\*\*\*\*\*\*\*\*\*\*\*\*\*\*\*\*\*\*\*\*\*\*\*\*\*\*\*\*\*\*\*\*\*

*Thirdly, I would like the authors to help the reader more in understanding the methodology. Figure 3 is helpful, but in the manuscript it is not always clear which data was used for what. Especially when explaining the equations on page 9 and 10, the authors could be clearer on which dataset was used, which time period. Also in the Results section it should be clarified when they are referring to calibration results, validation with POD data, or validation with the year 2017. For example, the first paragraph of Section 3.2.3 is quite confusing, because it discusses the performance of the models in the calibration period, which was already discussed in Section 3.2.1. Table 2 should be explained better; how is it different or similar to the information presented in Figures 7&8? Also, in the first paragraph of Section 3.3.2 the authors state that "the simulated RPoD fit well to RPoDONDE" (l.349), but wasn't that already discussed in Section 3.2.1 (Figure 7&8)?*

We modified the section 2.6 to better explain our methodology.

Datasets considered as inputs in the equations on pages 9 and 10 are successively the dataset 2011-2017 then the dataset 1989-2017. The non-exceedance frequency of discharge and groundwater levels F is computed at a daily time step leading to extrapolate daily values of RPoD. Parameters for each HER2-HR combination $a_1$; $b_1$ and $F_0$ for model LLR and $a_2$; $b_2$ for the LR model are successively determined by regression (Figure 3) using the calibration data between 2012 and 2016 of the two datasets (calibration period when ONDE observations are available over the whole year). The number of piezometers and gauging stations selected in each HER2-HR combination is different according to the datasets used as inputs (see section 2.4 and 2.5), leading to different values of F over the common period 2011-2017.

Table 2 shows the inter-annual NSE of both models with the two datasets as inputs while Figures 7 and 8 show the average NSE over the entire calibration period between 2012 and 2016. The values of NSE, during the year 2017, concern the validation period. The current Table 2 may be confusing and the calibration and validation NSEs have been specified in the revised paper:

| | | 2011-2017 dataset | | | | | | 1989-2017 dataset | | | | | |
|---|---|---|---|---|---|---|---|---|---|---|---|---|---|
| | | Calibration | | | | | Valid. | Calibration | | | | | Valid. |
| | | 2012 | 2013 | 2014 | 2015 | 2016 | **2017** | 2012 | 2013 | 2014 | 2015 | 2016 | **2017** |
| **LLR model** | May | 0.2 | 0.0 | 0.5 | 0.5 | 0.6 | **0.4** | 0.2 | 0.0 | 0.3 | 0.0 | 0.7 | **0.2** |
| | June | 0.6 | 0.3 | 0.8 | 0.5 | 0.8 | **0.5** | 0.6 | 0.3 | 0.5 | 0.3 | 0.8 | **0.5** |
| | July | 0.7 | 0.5 | 0.6 | 0.6 | 0.8 | **0.7** | 0.7 | 0.5 | 0.5 | 0.4 | 0.8 | **0.6** |
| | August | 0.8 | 0.6 | 0.7 | 0.7 | 0.8 | **0.6** | 0.7 | 0.5 | 0.5 | 0.5 | 0.8 | **0.6** |
| | Sept. | 0.7 | 0.8 | 0.6 | 0.6 | 0.7 | **0.6** | 0.6 | 0.7 | 0.5 | 0.5 | 0.6 | **0.6** |
| | May - Sept | 0.8 | 0.8 | 0.7 | 0.7 | 0.8 | **0.7** | 0.8 | 0.7 | 0.5 | 0.6 | 0.8 | **0.7** |
| **LR model** | May | 0.2 | 0.0 | 0.5 | 0.1 | 0.6 | **0.3** | 0.3 | 0.0 | 0.3 | 0.0 | 0.7 | **0.2** |
| | June | 0.6 | 0.5 | 0.8 | 0.5 | 0.8 | **0.4** | 0.6 | 0.4 | 0.5 | 0.3 | 0.7 | **0.4** |
| | July | 0.7 | 0.6 | 0.5 | 0.6 | 0.8 | **0.6** | 0.7 | 0.4 | 0.5 | 0.4 | 0.8 | **0.6** |
| | August | 0.7 | 0.6 | 0.7 | 0.6 | 0.7 | **0.6** | 0.6 | 0.4 | 0.5 | 0.4 | 0.7 | **0.5** |
| | Sept. | 0.6 | 0.8 | 0.6 | 0.7 | 0.7 | **0.6** | 0.5 | 0.6 | 0.4 | 0.5 | 0.6 | **0.6** |
| | May - Sept | 0.8 | 0.8 | 0.7 | 0.7 | 0.8 | **0.7** | 0.8 | 0.7 | 0.5 | 0.6 | 0.8 | **0.7** |

The first sentence of Section 3.2.3 is redundant with Section 3.2.1. We modified this paragraph to focus more on the annual performance of each model in the revised version.

The first paragraph of Section 3.3.2 briefly presents the model performance by graphically comparing simulated RPoDs with RPoD$_{ONDE}$. This paragraph only confirms the conclusions given above and have been shortened.
* * *
*Fourthly, it is unclear whether natural and/or human-influenced sites are selected in this study. In Section 2.4, the authors mention that the "observed discharges were not or only slightly altered by human actions" (p.7 l.164), but they do not specify whether the other datasets, i.e. groundwater levels, ONDE and POD observations, are near-natural too. This is important, as the authors mention in the discussion, "the basins are subject of intense agriculture with important water withdrawals during summer. Abstractions greatly reduce the water availability in rivers and in aquifers which are no longer able to support the low water levels and lead to increased flow intermittence. The responses of biological communities to artificial flow intermittence is still poorly understood compared to natural IRES." (p.19 l.435-439) If near-natural and human-influenced data are mixed in the predictions, it will be very difficult to understand the reasons for the regional patterns in drying and the statements about the highest drying occurring in sedimentary plains due to the low elevation gradient and dependence on rainfall might be flawed.*

We do agree that mixing natural and human-influenced stations bias the conclusions of the analysis and we modified the discussion (lines 487 to lines 499).

Here, the selection of the gauging stations inherits from previous studies and from the long expertise of the time series available in the HYDRO database. We have excluded stations with heavily modified river flow regime. As this selection is the result of expertise, we cannot be sure that there is absolutely no human action that may impact low flows.

The HYDRO database managers (section 2.4) consider as strongly influenced, gauging stations located on rivers regulated by dams, reservoirs or important water abstractions precisely localized, or on channelized rivers (e.g. diversion channel). As for the HYDRO gauging stations, ONDE sites are located on headwater streams without major human influence.

Regarding alteration issues in our datasets, we do not have access to the exact location and the volumes of water withdrawal for irrigation purposes. However, due to their upstream location, water availability is expected to be low, which may limit potential withdrawals and as consequence flow alteration at ONDE sites. Piezometers have been identified as involved in groundwater/surface water exchanges (section 2.5) and they experience seasonal fluctuations similar to the headwater streams monitored by the HYDRO database. The level of alteration of groundwater levels by water withdrawal is unknown because no information is available. However, in sedimentary plains where agricultural crops dominate the landscape, we are not sure that no human action affects low flows. Hopefully all the basins are not strongly affected by abstraction.

\*\*\*\*\*\*\*\*\*\*\*\*\*\*\*\*\*\*\*\*\*\*\*\*\*\*\*\*\*\*\*\*\*\*\*\*\*\*\*\*\*\*\*\*\*\*\*\*\*\*\*\*\*\*\*\*\*\*\*\*\*\*\*\*\*\*\*\*\*\*\*\*\*\*\*\*\*\*\*\*\*\*\*\*\*\*\*

*And finally, it is unclear why two statistical models are used throughout the paper. If they are equally suitable from a theoretical perspective, two (or more) models could be used for testing, but then the best model should be used to simulate the final results.*

Both models are equally suitable from a theoretical perspective and they demonstrate similar performance over the period 2012-2016. However, out of the calibration period (*i.e.* 1989-2011), both models are facing unexperienced climate conditions. As detailed in the last paragraph of the Discussion, the tails of the logarithmic curve and of the logistic curve are different and induce distinct values when the average of the non-exceedance frequencies *F* is close to 0%. As an illustration of the divergence of the models, maps of RPoD$_{LR}$ and RPoD$_{LLR}$ are displayed with *F* fixed to 1% in Figure 14. Predictions from the LLR model are thus larger than those from the LR model during generalized drought. We are not able to identify which model provides the more realistic values out of the conditions experienced over the calibration. Hence, we consider that presenting the results of these two models is of interest and keeping the two models is a way to put into perspective the estimated values - in particular those around the years 1989 to 1991 in response to extremely dry conditions.

\*\*\*\*\*\*\*\*\*\*\*\*\*\*\*\*\*\*\*\*\*\*\*\*\*\*\*\*\*\*\*\*\*\*\*\*\*\*\*\*\*\*\*\*\*\*\*\*\*\*\*\*\*\*\*\*\*\*\*\*\*\*\*\*\*\*\*\*\*\*\*\*\*\*\*\*\*\*\*\*\*\*\*\*\*\*\*

**Specific comments:**

*The regional probability of drying needs to be explained. In Section 2.6 the authors only mention that RPoD is calculated, but they never explain how this variable is calculated exactly.*

We added the definition of RPoD in section 2.2. Observed values of RPoD (RPoD$_{ONDE}$) is calculated as follows:

$$RPoD_{ONDE}(d) = \frac{(Ndrying)_{HER2-HR}}{(Nflowing + Ndrying)_{HER2-HR}}$$

where *d* denotes the observation date of the ONDE network, Ndrying and Nflowing are the number of drying and of flowing statuses observed at ONDE sites located in a same in a HER2-HR combination at the observation date *d*, respectively.

\*\*\*\*\*\*\*\*\*\*\*\*\*\*\*\*\*\*\*\*\*\*\*\*\*\*\*\*\*\*\*\*\*\*\*\*\*\*\*\*\*\*\*\*\*\*\*\*\*\*\*\*\*\*\*\*\*\*\*\*\*\*\*\*\*\*\*\*\*\*\*\*\*\*\*\*\*\*\*\*\*\*\*\*\*\*\*

*The weighted average of the non-exceedance frequencies (F) needs to be explained better. According to the Discussion section discharge and groundwater levels are combined (l.411-412), but this is not explained clearly enough in the Methods section (l.202-203). How are these non-exceedance frequencies of groundwater and discharge averaged since they have such different shapes and ranges (see Figure 3). And what do the authors mean with "with respect to the relative proportions of gauging stations and piezometers" (l.203-204)?*

We provided the details for computing F in the section 2.6.

F is computed for each HER2-HR combination:

Let us consider a day (d) and the gauging stations and piezometers available in the HER2-HR combination.

The non-exceedance frequency of the discharge observed at the day *d*, Fq, is determined for each gauging station using the flow duration curve. In the same way, the non-exceedance frequencies of the groundwater levels Fgw observed the same day is determined for each piezometer.

The average of the non-exceedance frequencies (F) is calculated following the next equation:

$$F(d) = \frac{\sum_{i=1}^{i=Nq} Fq_i}{Nq} \times \frac{Nq}{(Nq + Ngw)} + \frac{\sum_{j=1}^{j=Ngw} Fgw_j}{Ngw} \times \frac{Ngw}{(Nq + Ngw)} = \frac{\sum_{i=1}^{i=Nq} Fq_i + \sum_{j=1}^{j=Ngw} Fgw_j}{(Nq + Ngw)}$$

with $Fq_i$: the mean non-exceedance frequency of discharge at the gauging station i calculated between d and d-5; $Fgw_j$: the mean non-exceedance frequency of groundwater levels at the piezometer j calculated between d and d-5; Nq: the number of gauging stations selected in a HER2-HR combination and Ngw: the number of selected piezometers selected in the HER2-HR combination. The non-exceedance frequency combining discharge and groundwater levels characterize a general hydrological state at a HER2-HR scale.

\*\*\*\*\*\*\*\*\*\*\*\*\*\*\*\*\*\*\*\*\*\*\*\*\*\*\*\*\*\*\*\*\*\*\*\*\*\*\*\*\*\*\*\*\*\*\*\*\*\*\*\*\*\*\*\*\*\*\*\*\*\*\*\*\*\*\*\*\*\*\*\*\*\*\*\*\*\*\*\*\*\*\*\*\*\*\*

*The authors conclude that "both models seem able to predict RPoD out of the calibration period" (l. 330-331), but do a NSE of 0.4 and 0.5 warrant such a statement?*

This section (section 3.2.3) has been modified and the revised manuscript presents NSEs for the 2017 validation year. Table 2 has been modified and presents these additional results. Figure 10 has been modified and shows the dispersion between predicted RPoD and drying observed at ONDE sites in the scatter plot during the validation year 2017 (Fig. 10a and 10b) in comparison with the year 2012 which obtains the better NSE during calibration period (Fig. 10c and 10d). The NSE obtain in 2017 are 0.72 with the LLR model and 0.68 with the LR model against respectively 0.83 and 0.81 in 2012. The

performance is slightly lower in 2017 but remains acceptable with NSEs close to 0.7 and both models seem able to predict RPoD out of the calibration period.

[Figure]

**Figure 10.** Scatter plot of the predicted RPoD (x axis) and drying observed at ONDE sites (y axis) in 2017 and 2012 simulated with the 2011-2017 dataset by: (a) and (c) the LLR model and (b) and (d) the LR model.

\*\*\*\*\*\*\*\*\*\*\*\*\*\*\*\*\*\*\*\*\*\*\*\*\*\*\*\*\*\*\*\*\*\*\*\*\*\*\*\*\*\*\*\*\*\*\*\*\*\*\*\*\*\*\*\*\*\*\*\*\*\*\*\*\*\*\*\*\*\*\*\*\*\*\*\*\*\*\*\*

*A significant part of the Conclusion section discusses future work. Is that relevant for this manuscript? I would suggest leaving those paragraphs out as they distract from the main message of this paper.*

The authors wanted to highlight the perspectives to this work and to show the possible ways to predict RPoD at the local scale. The authors have taken this remark into account and shortened this part of the conclusion.

\*\*\*\*\*\*\*\*\*\*\*\*\*\*\*\*\*\*\*\*\*\*\*\*\*\*\*\*\*\*\*\*\*\*\*\*\*\*\*\*\*\*\*\*\*\*\*\*\*\*\*\*\*\*\*\*\*\*\*\*\*\*\*\*\*\*\*\*\*\*\*\*\*\*\*\*\*\*\*\*

**Textual comments:**

Thank you for your very attentive reading, all your corrections/suggestions have been taken into account. We also took into account the remark about the concept of RPoD which will be better

detailed and we will present the equation to compute the values of RPoD$_{ONDE}$ (Eq. 1; Page 6, L140-145). This formula can also be applied to derive the values of RPoD$_{POC}$ (Page 7, L168-170).

**References**

Buytaert, W., Zulkafli, Z., Grainger, S., Acosta, L., Alemie, T. C., Bastiaensen, J., De Bi??vre, B., Bhusal, J., Clark, J., Dewulf, A., Foggin, M., Hannah, D. M., Hergarten, C., Isaeva, A., Karpouzoglou, T., Pandeya, B., Paudel, D., Sharma, K., Steenhuis, T., Tilahun, S., Van Hecken, G. and Zhumanova, M.: Citizen science in hydrology and water resources: opportunities for knowledge generation, ecosystem service management, and sustainable development, Frontiers in Earth Science, 2, doi:10.3389/feart.2014.00026, 2014.

D'Ambrosio, E., De Girolamo, A. M., Barca, E., Ielpo, P. and Rulli, M. C.: Characterising the hydrological regime of an ungauged temporary river system: a case study, Environmental Science and Pollution Research, 24(16), 13950–13966, doi:10.1007/s11356-016-7169-0, 2017.

De Girolamo, A. M., Barca, E., Pappagallo, G. and Lo Porto, A.: Simulating ecologically relevant hydrological indicators in a temporary river system, Agricultural Water Management, 180, 194–204, doi:10.1016/j.agwat.2016.05.034, 2017.

Gallart, F., Llorens, P., Latron, J., Cid, N., Rieradevall, M. and Prat, N.: Validating alternative methodologies to estimate the regime of temporary rivers when flow data are unavailable, Science of The Total Environment, 565, 1001–1010, doi:10.1016/j.scitotenv.2016.05.116, 2016.

Kelso, J. E. and Entrekin, S. A.: Intermittent and perennial macroinvertebrate communities had similar richness but differed in species trait composition depending on flow duration, Hydrobiologia, 807(1), 189–206, doi:10.1007/s10750-017-3393-y, 2018.

Nadeau, T.-L. and Rains, M. C.: Hydrological Connectivity Between Headwater Streams and Downstream Waters: How Science Can Inform Policy1: Hydrological Connectivity Between Headwater Streams and Downstream Waters: How Science Can Inform Policy, JAWRA Journal of the American Water Resources Association, 43(1), 118–133, doi:10.1111/j.1752-1688.2007.00010.x, 2007.

Olden, J. D. and Poff, N. L.: Redundancy and the choice of hydrologic indices for characterizing streamflow regimes, River Research and Applications, 19(2), 101–121, doi:10.1002/rra.700, 2003.

Poff, N. L. and Ward, J. V.: Physical Habitat Template of Lotie Systems: Recovery in the Context of Historical Pattern of Spatiotemporal Heterogeneity, Environmental Management EMNGDC, 14(5), 1990.

Richter, B. D., Baumgartner, J. V., Powell, J. and Braun, D. P.: A method for assessing hydrologic alteration within ecosystems, Conservation biology, 10(4), 1163–1174, 1996.

Snelder, T. H., Lamouroux, N., Leathwick, J. R., Pella, H., Sauquet, E. and Shankar, U.: Predictive mapping of the natural flow regimes of France, Journal of Hydrology, 373(1-2), 57–67, doi:10.1016/j.jhydrol.2009.04.011, 2009.

Snelder, T. H., Datry, T., Lamouroux, N., Larned, S. T., Sauquet, E., Pella, H. and Catalogne, C.: Regionalization of patterns of flow intermittence from gauging station records, Hydrology and Earth System Sciences, 17(7), 2685–2699, doi:10.5194/hess-17-2685-2013, 2013.

Turner, D. S. and Richter, H. E.: Wet/Dry Mapping: Using Citizen Scientists to Monitor the Extent of Perennial Surface Flow in Dryland Regions, Environmental Management, 47(3), 497–505, doi:10.1007/s00267-010-9607-y, 2011.

---

## Author Comment (AC2) · 6 Mar 2018

**Reply to Catherine Sefton**

*This paper makes a valuable contribution to the study of the hydrology of intermittent rivers and ephemeral streams. In particular, it is the first time observations of river flow and groundwater level have been used in combination with observations of hydrological state in a regionalisation approach, marking a step forward in the modelling and mapping of intermittency at national scale.*

*The ONDE dataset is unique in the literature, notably the large number of sites, the coverage of headwater streams and the national extent, but has limitations in the summer-only timing and small number of observations. The merging of the "no visible flow" status with the "dried out" status means that a key benefit of the dataset is not utilised, as the two-status classification of flowing and drying that remains is no advantage over that available from gauged river flow data. Discussion of the network would benefit from broader contextual comment on the contribution and application of these data.*

*The paper is well written and referenced and is recommended for final publication with minor revisions.*

The authors would like to thank Catherine Sefton for her positive evaluation of our paper and the specific comments and text/figure corrections that will lead to improve the manuscript. The detailed answers to the specific comments are presented below.

\*\*\*\*\*\*\*\*\*\*\*\*\*\*\*\*\*\*\*\*\*\*\*\*\*\*\*\*\*\*\*\*\*\*\*\*\*\*\*\*\*\*\*\*\*\*\*\*\*\*\*\*\*\*\*\*\*\*\*\*\*\*\*\*\*\*\*\*\*\*\*\*\*\*\*\*\*\*\*\*\*\*\*\*\*\*\*\*\*\*\*\*\*\*\*\*\*\*

**Specific comments**

*Given the small number of observations at each site, the claim that the ONDE dataset offers more accurate assessment of inter-annual variability than the gauging station network (L376-377) needs further justification. Conversely, the claim that the dataset makes it possible to capture drying events at the regional scale (L381-382) would benefit from stressing the monitoring of both upstream and downstream drying – uniquely each with national extent – in your approach.*

With this statement, the authors wanted to highlight the added information provided by the ONDE network in comparison with the HYDRO database. The location of these observation sites on headwater streams is more adapted to identify IRES along the river network (about 2 400 ONDE sites have a drainage area < 50 km² against 850 gauging stations in the HYDRO database). Despite the limited number of observations available at each ONDE site, the regional approach developed here has succeeded in reconstructing drying dynamics at the daily time step and this reconstruction allows thereafter examining the inter-annual variability of drying occurrence. This study cannot be performed at the regional scale using the set of gauging stations due to the low proportion of gauged head streams in the HYDRO database.

In section 3.1.2, the drying detection by the ONDE network is compared to HYDRO database. In Table 1, we show that the frequency of drying for IRES available in the HYDRO database does not vary much from one year to another (about 30% whether in wet year or in dry year). This variability seems low compared to the drying frequencies observed each year with the ONDE network (Fig. 4b and

Fig. 5). To illustrate this, we added above the figure of the distribution of the percentages of drying observed at gauging stations from HYDRO database for each year following the same symbology than the figure 5 in the manuscript. The percentages of drying are very similar during wet years (2013, 2014) and dry years (2012, 2015, 2016). In that sense, we wanted to underline the interest of using the ONDE network to improve our knowledge on the temporal pattern of the frequency of drying.

[Figure]

○ 0 to 25%   ○ 25% to 50%   ⬤ 50 to 75%   ⬤ 75 to 100%

**Figure 1bis.** Distribution of the percentages of drying observed at gauging stations HYDRO for the years: (a) 2012, (b) 2013, (c) 2014, (d) 2015, and (e) 2016.

Estimates are only valid at the regional scale. Unfortunately the approach is unable to provide information about how the dry events develop in space (patch connectivity). The method does not use any information about upstream-downstream dependencies that would be required for example for mapping purposes.
* * *
*The presentation of summer-only status data as "% drying" needs qualification, as it suggests assumptions about the status in the rest of the year. In particular, clarification would be helpful in line 242, when the context implies it means the number of sites with at least one drying each year (as in line 241), rather than the % of all observations at all sites (as in Fig 4).*

We modified these sentences in the revised paper (Lines 271-275 and Lines 443-447).

The first sentence presents the number of sites where, at least, one drying event is observed over the period 2012-2016, to identify the headwater streams in the ONDE network that can be considered as

IRES. At the end of the first paragraph of section 3.3.1, the proportion of drying over France was computed as the total number of drying observed with the ONDE network over France divided by the total number of ONDE observations available during the same year (Fig. 4a).

\*\*\*\*\*\*\*\*\*\*\*\*\*\*\*\*\*\*\*\*\*\*\*\*\*\*\*\*\*\*\*\*\*\*\*\*\*\*\*\*\*\*\*\*\*\*\*\*\*\*\*\*\*\*\*\*\*\*\*\*\*\*\*\*\*\*\*\*\*\*\*\*\*\*\*\*\*\*\*\*\*\*\*\*\*\*\*\*

*The technique of constructing mean non-exceedance frequency from river flows and groundwater level is attractive and robust. However, its limitation in this regional approach of failing to capture the effect of local rainfall should be commented upon, especially given the dominance of rainfall-driven intermittency stated in section 3.3.2.*

This is discussed in the revised paper (Lines 475-481).

The factors involved in the in-situ drying dynamics are numerous. Rainfall is one of these factors and has a significant effect on the re-wetted streams during rainfall convective episodes. The authors agree that the mean non-exceedance frequency is a global index that only captures the hydrological conditions at the regional scale in modelling the RPoD. However we may expect that for rainfall-driven river flow regime, the effect of rainfall events on flow intermittence at the HER2-HR scale is contained in the daily discharges and groundwater levels used to the mean non-exceedance frequency and that the effect of rain on the proportion of drying is indirectly taken into account.

In the case of a local rainfall event that affects an ungauged basin, we may miss key information to simulate the possible end of the drought of the affected region. This is one of the limits of the regional approach.

\*\*\*\*\*\*\*\*\*\*\*\*\*\*\*\*\*\*\*\*\*\*\*\*\*\*\*\*\*\*\*\*\*\*\*\*\*\*\*\*\*\*\*\*\*\*\*\*\*\*\*\*\*\*\*\*\*\*\*\*\*\*\*\*\*\*\*\*\*\*\*\*\*\*\*\*\*\*\*\*\*\*\*\*\*\*\*\*

*The frequency of drying from gauging station data needs to be defined (line 272). Context suggests it is flow permanence (dry days or dry months per time period), but frequency in intermittent rivers and ephemeral streams can also mean dry spells per time period, and it also needs to be clear and justified whether it's calculated from daily means or monthly means.*

Indeed the sentence is confusing. The frequency of drying described here corresponds to the ratio between the number of dry days and the total number of days between the 1[st] May and the 30[th] September of one year (273-121+1= 153 days).

\*\*\*\*\*\*\*\*\*\*\*\*\*\*\*\*\*\*\*\*\*\*\*\*\*\*\*\*\*\*\*\*\*\*\*\*\*\*\*\*\*\*\*\*\*\*\*\*\*\*\*\*\*\*\*\*\*\*\*\*\*\*\*\*\*\*\*\*\*\*\*\*\*\*\*\*\*\*\*\*\*\*\*\*\*\*\*\*

*In section 3.2.1., the difference in performance between the two explanatory hydrological datasets is attributed to the difference in the number of gauging stations and piezometers. The pattern in Figure 8 is not as clear as the text suggests, and it would be good to comment also on the assumption of stationarity and how it might vary between HER2-HR combinations. Similarly, historical reconstructions make assumptions about stationarity that need to be acknowledged.*

The question on stationarity arises due to uncomplete information about the applications to the two datasets. Models are calibrated against observation available during the same period (i.e. 2012-2016). However the selected piezometers and gauging stations differ according to the dataset resulting in different time series of mean non exceedance frequency representative of the period

2012-2016. Thus there are two sets of parameters specific to each dataset (see section 2.4 and 2.5) for both LLR and LR models.

We revised the description of the applications (Lines 258-260).

\*\*\*\*\*\*\*\*\*\*\*\*\*\*\*\*\*\*\*\*\*\*\*\*\*\*\*\*\*\*\*\*\*\*\*\*\*\*\*\*\*\*\*\*\*\*\*\*\*\*\*\*\*\*\*\*\*\*\*\*\*\*\*\*\*\*\*\*\*\*\*\*\*\*\*\*\*\*\*\*\*\*

*The conclusion is a good summary of the results but would benefit from contextual comment, both with respect to the stated objective of this paper and more broadly on the contribution being made to the field.*

We modified the conclusion in the revised paper in order to highlight the contribution of our study (Lines 544-551).

\*\*\*\*\*\*\*\*\*\*\*\*\*\*\*\*\*\*\*\*\*\*\*\*\*\*\*\*\*\*\*\*\*\*\*\*\*\*\*\*\*\*\*\*\*\*\*\*\*\*\*\*\*\*\*\*\*\*\*\*\*\*\*\*\*\*\*\*\*\*\*\*\*\*\*\*\*\*\*\*\*\*

**Textual and Figure corrections:**

Thank you for your very attentive reading, all your corrections/suggestion will be taken into account.

*Figure 3, step 1 : It is unclear why HR1, 4 and 6 are shown as types of monitoring site, when section 2.1 has defined them as types of hydrological regime.*

HR means "hydrological regime" and the figure 3 shows one HER2 (HER2 n°97) which contains streams which have 3 types of hydrological regime (HR1, HR4 and HR6). For the example shown on the figure, all ONDE sites located on streams with a HR with a type 6 are selected in order to make the regression. We clarified this aspect on a revised Figure 3.

*Figure 10: This would benefit from additional plots for a year that has good NSE, as the text is comparing years as well as model performance.*

We added additional plots of the year 2012 which obtain the better NSE during the calibration period and we added observations *vs.* predictions of the full year 2017 in a revised Figure 10. The section 3.2.3 and the Table 2 have been revised in order to present these new results.

---

## Author Response (AR1)

**Reply to Anne Van Loon**

**General comments**

*Firstly, the authors need to explain why a regional assessment of headwater drying is needed. What is the benefit of Figure 11 over Figure 5? The patterns of drying are the same, so Figure 5 would be sufficient to indicate hotspots of drying within France and temporal variability in drying.*

We modified lines 88-94 to better explain that one of our objective is a temporal extrapolation of the daily drying probability (eg Fig 11) in regions, based on discrete observations (~5/years, raw data in Fig 5).
* * *
*Secondly, the paper is focused on France. This in itself is not a problem, since the methodology and results are interesting and useful beyond France, but the author fail to put their findings in a broader perspective in the discussion. Literature on IRES research from outside France should be discussed and the authors should clarify what is new and interesting about this work from an international perspective. On p.19 l.443-452, the authors mention how their results are consistent with previous studies, which is great, but they should additionally point out what their study adds. If this is not done, the study would be better placed in a Journal like Journal of Hydrology – Regional Studies.*

The authors agree with this remark. We modified the text (lines 465-481 and lines 544-551) to better explain the international relevance of our results.
* * *
*Thirdly, I would like the authors to help the reader more in understanding the methodology. Figure 3 is helpful, but in the manuscript it is not always clear which data was used for what. Especially when explaining the equations on page 9 and 10, the authors could be clearer on which dataset was used, which time period. Also in the Results section it should be clarified when they are referring to calibration results, validation with POD data, or validation with the year 2017. For example, the first paragraph of Section 3.2.3 is quite confusing, because it discusses the performance of the models in the calibration period, which was already discussed in Section 3.2.1. Table 2 should be explained better; how is it different or similar to the information presented in Figures 7&8? Also, in the first paragraph of Section 3.3.2 the authors state that "the simulated RPoD fit well to RPoDONDE" (l.349), but wasn't that already discussed in Section 3.2.1 (Figure 7&8)?*

We modified the section 2.6 to better explain our methodology. The table 2 has been revised.

We modified the Section 3.2.3 to focus on the annual performance of each model in the revised manuscript.
* * *
*Fourthly, it is unclear whether natural and/or human-influenced sites are selected in this study. In Section 2.4, the authors mention that the "observed discharges were not or only slightly altered by human actions" (p.7 l.164), but they do not specify whether the other datasets, i.e. groundwater levels, ONDE and POD observations, are near-natural too. This is important, as the authors mention in the discussion, "the basins are subject of intense agriculture with important water withdrawals during summer. Abstractions greatly reduce the water availability in rivers and in aquifers which are no longer able to support the low water levels and lead to increased flow intermittence. The responses of biological communities to artificial flow intermittence is still poorly understood compared to natural IRES." (p.19 l.435-439) If near-natural and human-influenced data are mixed in the predictions, it will be very difficult to understand the reasons for the regional patterns in drying and the statements about the highest drying occurring in sedimentary plains due to the low elevation gradient and dependence on rainfall might be flawed.*

==We modified the discussion (lines 487 to lines 499).==

*\*\*\*\*\*\*\*\*\*\*\*\*\*\*\*\*\*\*\*\*\*\*\*\*\*\*\*\*\*\*\*\*\*\*\*\*\*\*\*\*\*\*\*\*\*\*\*\*\*\*\*\*\*\*\*\*\*\*\*\*\*\*\*\*\*\*\*\*\*\*\*\*\*\*\*\*\*\*\*\*\*\**

*And finally, it is unclear why two statistical models are used throughout the paper. If they are equally suitable from a theoretical perspective, two (or more) models could be used for testing, but then the best model should be used to simulate the final results.*

==We modified the discussion (lines 517 to lines 522).==

\*\*\*\*\*\*\*\*\*\*\*\*\*\*\*\*\*\*\*\*\*\*\*\*\*\*\*\*\*\*\*\*\*\*\*\*\*\*\*\*\*\*\*\*\*\*\*\*\*\*\*\*\*\*\*\*\*\*\*\*\*\*\*\*\*\*\*\*\*\*\*\*\*\*\*\*\*\*\*\*\*\*\*

**Specific comments:**

*The regional probability of drying needs to be explained. In Section 2.6 the authors only mention that RPoD is calculated, but they never explain how this variable is calculated exactly.*

==We added the definition of RPoD in section 2.2.==
\*\*\*\*\*\*\*\*\*\*\*\*\*\*\*\*\*\*\*\*\*\*\*\*\*\*\*\*\*\*\*\*\*\*\*\*\*\*\*\*\*\*\*\*\*\*\*\*\*\*\*\*\*\*\*\*\*\*\*\*\*\*\*\*\*\*\*\*\*\*\*\*\*\*\*\*\*\*\*\*\*\*\*

*The weighted average of the non-exceedance frequencies (F) needs to be explained better. According to the Discussion section discharge and groundwater levels are combined (l.411-412), but this is not explained clearly enough in the Methods section (l.202-203). How are these non-exceedance frequencies of groundwater and discharge averaged since they have such different shapes and ranges (see Figure 3). And what do the authors mean with "with respect to the relative proportions of gauging stations and piezometers" (l.203-204)?*

==We provided the details to better explain F in the section 2.6.==

\*\*\*\*\*\*\*\*\*\*\*\*\*\*\*\*\*\*\*\*\*\*\*\*\*\*\*\*\*\*\*\*\*\*\*\*\*\*\*\*\*\*\*\*\*\*\*\*\*\*\*\*\*\*\*\*\*\*\*\*\*\*\*\*\*\*\*\*\*\*\*\*\*\*\*\*\*\*\*\*\*\*

*The authors conclude that "both models seem able to predict RPoD out of the calibration period" (l. 330-331), but do a NSE of 0.4 and 0.5 warrant such a statement?*

==This section (section 3.2.3) has been modified and the revised manuscript presents NSEs for the 2017 validation year. Table 2 has been modified and presents these additional results. Figure 10 has been==

modified and shows the dispersion between predicted RPoD and drying observed at ONDE sites in the scatter plot during the validation year 2017 (Fig. 10a and 10b) in comparison with the year 2012 which obtains the better NSE during calibration period (Fig. 10c and 10d).

\*\*\*\*\*\*\*\*\*\*\*\*\*\*\*\*\*\*\*\*\*\*\*\*\*\*\*\*\*\*\*\*\*\*\*\*\*\*\*\*\*\*\*\*\*\*\*\*\*\*\*\*\*\*\*\*\*\*\*\*\*\*\*\*\*\*\*\*\*\*\*\*\*\*\*\*\*\*\*\*\*\*\*\*\*\*\*\*\*\*\*

*A significant part of the Conclusion section discusses future work. Is that relevant for this manuscript? I would suggest leaving those paragraphs out as they distract from the main message of this paper.*

The authors have shortened this part of the conclusion.

\*\*\*\*\*\*\*\*\*\*\*\*\*\*\*\*\*\*\*\*\*\*\*\*\*\*\*\*\*\*\*\*\*\*\*\*\*\*\*\*\*\*\*\*\*\*\*\*\*\*\*\*\*\*\*\*\*\*\*\*\*\*\*\*\*\*\*\*\*\*\*\*\*\*\*\*\*\*\*\*\*\*\*\*\*\*\*\*\*\*\*

**Textual comments:**

All your corrections/suggestions have been taken into account. We also took into account the remark about the concept of RPoD which will be better detailed and we will present the equation to compute the values of $RPoD_{ONDE}$ (Eq. 1; Page 6, L140-145). This formula can also be applied to derive the values of $RPoD_{POC}$ (Page 7, L168-170).

**Reply to Catherine Sefton**

**Specific comments**

*Given the small number of observations at each site, the claim that the ONDE dataset offers more accurate assessment of inter-annual variability than the gauging station network (L376-377) needs further justification. Conversely, the claim that the dataset makes it possible to capture drying events at the regional scale (L381-382) would benefit from stressing the monitoring of both upstream and downstream drying – uniquely each with national extent – in your approach.*

\*\*\*\*\*\*\*\*\*\*\*\*\*\*\*\*\*\*\*\*\*\*\*\*\*\*\*\*\*\*\*\*\*\*\*\*\*\*\*\*\*\*\*\*\*\*\*\*\*\*\*\*\*\*\*\*\*\*\*\*\*\*\*\*\*\*\*\*\*\*\*\*\*\*\*\*\*\*\*\*\*\*\*\*\*\*

*The presentation of summer-only status data as "% drying" needs qualification, as it suggests assumptions about the status in the rest of the year. In particular, clarification would be helpful in line 242, when the context implies it means the number of sites with at least one drying each year (as in line 241), rather than the % of all observations at all sites (as in Fig 4).*

We modified these sentences in the revised paper (Lines 271-275 and Lines 443-447).

\*\*\*\*\*\*\*\*\*\*\*\*\*\*\*\*\*\*\*\*\*\*\*\*\*\*\*\*\*\*\*\*\*\*\*\*\*\*\*\*\*\*\*\*\*\*\*\*\*\*\*\*\*\*\*\*\*\*\*\*\*\*\*\*\*\*\*\*\*\*\*\*\*\*\*\*\*\*\*\*\*\*\*\*\*\*

*The technique of constructing mean non-exceedance frequency from river flows and groundwater level is attractive and robust. However, its limitation in this regional approach of failing to capture the effect of local rainfall should be commented upon, especially given the dominance of rainfall-driven intermittency stated in section 3.3.2.*

This is discussed in the revised paper (Lines 475-481).

\*\*\*\*\*\*\*\*\*\*\*\*\*\*\*\*\*\*\*\*\*\*\*\*\*\*\*\*\*\*\*\*\*\*\*\*\*\*\*\*\*\*\*\*\*\*\*\*\*\*\*\*\*\*\*\*\*\*\*\*\*\*\*\*\*\*\*\*\*\*\*\*\*\*\*\*\*\*\*\*\*\*\*\*\*

*The frequency of drying from gauging station data needs to be defined (line 272). Context suggests it is flow permanence (dry days or dry months per time period), but frequency in intermittent rivers and ephemeral streams can also mean dry spells per time period, and it also needs to be clear and justified whether it's calculated from daily means or monthly means.*

We revised this section (Lines 304-306).

\*\*\*\*\*\*\*\*\*\*\*\*\*\*\*\*\*\*\*\*\*\*\*\*\*\*\*\*\*\*\*\*\*\*\*\*\*\*\*\*\*\*\*\*\*\*\*\*\*\*\*\*\*\*\*\*\*\*\*\*\*\*\*\*\*\*\*\*\*\*\*\*\*\*\*\*\*\*\*\*\*\*\*\*\*

*In section 3.2.1., the difference in performance between the two explanatory hydrological datasets is attributed to the difference in the number of gauging stations and piezometers. The pattern in Figure 8 is not as clear as the text suggests, and it would be good to comment also on the assumption of stationarity and how it might vary between HER2-HR combinations. Similarly, historical reconstructions make assumptions about stationarity that need to be acknowledged.*

We revised the description of the applications (Lines 237-244).

\*\*\*\*\*\*\*\*\*\*\*\*\*\*\*\*\*\*\*\*\*\*\*\*\*\*\*\*\*\*\*\*\*\*\*\*\*\*\*\*\*\*\*\*\*\*\*\*\*\*\*\*\*\*\*\*\*\*\*\*\*\*\*\*\*\*\*\*\*\*\*\*\*\*\*\*\*\*\*\*\*\*\*\*\*

*The conclusion is a good summary of the results but would benefit from contextual comment, both with respect to the stated objective of this paper and more broadly on the contribution being made to the field.*

We modified the conclusion in the revised paper in order to highlight the contribution of our study (Lines 544-551).

\*\*\*\*\*\*\*\*\*\*\*\*\*\*\*\*\*\*\*\*\*\*\*\*\*\*\*\*\*\*\*\*\*\*\*\*\*\*\*\*\*\*\*\*\*\*\*\*\*\*\*\*\*\*\*\*\*\*\*\*\*\*\*\*\*\*\*\*\*\*\*\*\*\*\*\*\*\*\*\*\*\*\*\*

**Textual and Figure corrections:**

*Figure 3, step 1 : It is unclear why HR1, 4 and 6 are shown as types of monitoring site, when section 2.1 has defined them as types of hydrological regime.*

We clarified this aspect on a revised Figure 3.

*Figure 10: This would benefit from additional plots for a year that has good NSE, as the text is comparing years as well as model performance.*

We added additional plots of the year 2012 which obtain the better NSE during the calibration period and we added observations *vs.* predictions of the full year 2017 in a revised Figure 10. The section 3.2.3 and the Table 2 have been revised in order to present these new results.

[revised manuscript text omitted]

NSE calculated with the 2011-2017 dataset:

a     b

125  250 Km

☐ No data ☐ 0 - 0.2 ▨ 0.2 - 0.4 ▨ 0.4 - 0.6 ▨ 0.6 - 0.8 ■ 0.8 - 1

NSE calculated with the 1989-2017 dataset:

c     d

125  250 Km

☐ No data ☐ 0 - 0.2 ▨ 0.2 - 0.4 ▨ 0.4 - 0.6 ▨ 0.6 - 0.8 ■ 0.8 - 1

(NSE 2011-2017 dataset - NSE 1989-2017 dataset):

e     f

125  250 Km

☐ 0 ▨ 0 to 0.1 ▨ 0.1 to 0.2 ■ 0.2 to 0.3

**Figure 7.** Map of Nash-Sutcliffe criteria (NSE) obtained for each HER2-HR combination between 2012
and 2016 with the 2011-2017 and 1989-2017 datasets according to: (a) and (c) a log-linear regression
(LLR) model; (b) and (d) a logistic regression (LR) model. NSE differences between the 2011-2017
dataset and the 1989-2017 dataset are represented for: (e) LLR model and (f) LR model.

[Figure]

**Figure 8.** NSE calculated for each HER2-HR combination between 2012 and 2016 with the 1989-2017
dataset as a function of NSE calculated with 2011-2017 dataset with respectively: (a) the LLR model
and (b) the LR model. The color of dots represents the proportion of gauging station and piezometers
lost between the 2011-2017 database and the 1989-2017 database: losses < 50% (white); losses
between 50% and 75% (grey); losses > 75% (black).

[Figure]

**RPoD simulated with the 2011-2017 dataset:**

**RPoD simulated with the 1989-2017 dataset:**

LLR model --- LR model ○ Obs - POC data

**Figure 9.** Comparison between observed proportion of drying RPoD$_{POC}$ and RPoD predicted by the LLR
and LR models with the 2011-2017 dataset in: (a) 2011, (b) 2012 (c) 2013 and with the 1989-2017
dataset in: (d) 2011, (e) 2012 (f) 2013.

[Figure]

**Figure 10.** Scatter plot of the predicted RPoD (x axis) and drying observed at ONDE sites (y axis) in
2017 and 2012 simulated with the 2011-2017 dataset by: (a) and (c) the LLR model and (b) and (d)
the LR model.

[Figure]

**Figure 11.** Maximum duration of consecutive days with RPoD higher than 20% simulated with LLR
and LR model.

[Figure]

**Figure 12.** RPoD simulated between 1989 and 2016 the 1989-2017 dataset with: (a) the LR model and
(b) the LLR model. The grey area represents the RPoD between the 90[th] percentile and the 10[th]
percentile simulated on HER2-HR combination, the black curve represents the average RPoD
simulated by HER2-HR combination and white dots represent the mean $RPoD_{ONDE}$ for each
observation dates. Dates mentioned correspond to the day of the maximum average RPoD simulated
by HER2-HR combination (black curve) of each year.

[Figure]

Figure 13. Comparison of NSE obtained with regression including only discharge variable as a function of NSE obtained with including discharge and groundwater level variables in the 2011-2017 dataset with: (a) LLR model and (b) LR model.

[Figure]

**Figure 14.** Regional probability of drying simulated with F = 1% predicted with: (a) the LLR model and (b) the LR model.